# Glacier surge monitoring from temporally dense elevation time series: application to an ASTER dataset over the Karakoram region

Luc Beraud<sup>1</sup>, Fanny Brun<sup>1</sup>, Amaury Dehecq<sup>1</sup>, Romain Hugonnet<sup>2</sup>, and Prashant Shekhar<sup>3</sup>

<sup>1</sup>Université Grenoble Alpes, CNRS, IRD, Grenoble INP, INRAE, Institut des Géosciences de l'Environnement (IGE), 38000 Grenoble. France

Correspondence: Luc Beraud (luc.beraud@univ-grenoble-alpes.fr, luc.beraud@protonmail.com)

#### Abstract.

Glacier surges are spectacular events that lead to surface elevation changes of tens of metres in a period of a few months to a few years, with different patterns of mass transport. Existing methods to derive elevation change associated with surges, and subsequent quantification of the transported mass, rely on differencing pairs of digital elevation models (DEMs) that may not be acquired regularly in time. In this study, we propose a workflow to filter and interpolate a dense time series of DEMs specifically for the study of surge events. We test this workflow on a global 20-year dataset of DEMs from the optical satellite sensor Advanced Spaceborne Thermal Emission and Reflection Radiometer (ASTER). The multistep procedure includes linear non-parametric Locally Weighted Regression and Smoothing Scatterplots (LOWESS) filtering and Approximation by Localized Penalized Splines (ALPS) interpolation. We run the workflow over the Karakoram mountain range (High Mountain Asia). We compare the produced dataset to previous studies for four selected surge events, on Hispar, Khurdopin, Kyagar and Yazghil glaciers. We demonstrate that our workflow captures thickness changes on a monthly scale with detailed patterns of mass transportation. Such patterns include surge front propagation and changes in dynamic balance line, among others. Our results allow a remarkably detailed description of glacier surges at the scale of a large region. The workflow preserves most of the elevation change signal, with underestimation or smoothing in a limited number of surge cases.

## 5 1 Introduction

Surge events are extreme cases of the continuous spectrum of glacier flow instabilities (Herreid and Truffer, 2016). Surges are quasi-periodic events characterized by abnormally rapid glacier flow, lasting from several months to years (Bhambri et al., 2017; Cuffey and Paterson, 2010). Large masses of ice are transported during surge events, causing thickness changes (Bhambri et al., 2017, 2022). They occur on a limited number of glaciers known as "surge-type" glaciers, which are clustered in a few regions of the globe, among which the Karakoram in High Mountain Asia (Guillet et al., 2022; Sevestre and Benn, 2015). Surges can occur on both land-terminating and tidewater glaciers, and on either polythermal or temperate glaciers (Cuffey and Paterson, 2010). The mechanisms behind the surge phenomenon (origin, surge trigger, etc.) are not yet fully understood and are an ongoing focus of theoretical investigations (e.g., Benn et al., 2023; Crompton et al., 2018; Terleth et al., 2021; Thøgersen

<sup>&</sup>lt;sup>2</sup>University of Washington, Civil and Environmental Engineering, Seattle, WA, USA

<sup>&</sup>lt;sup>3</sup>Embry-Riddle Aeronautical University, Daytona Beach Campus, FL, USA

50

Observations of glacier surface elevation change over time are extremely useful to document glacier surges, and can give insight into the current state of a glacier in its surge cycle. The surge period, active phase of the surge-type glacier, is characterised by thinning (i.e. decrease of surface elevation) in a reservoir area and thickening in a receiving area, representative of the ongoing mass transfer. The quiescence period consists in strong thinning in the receiving area of the previous surge, and a thickness increase (mass build-up) before the next surge and mostly in the future reservoir area. The differencing of elevation maps permits one to compute the volume of ice transferred during a surge event and determine the spatial extent affected (e.g., Bhambri et al., 2022; Gao et al., 2024; Steiner et al., 2018). A few surge-type glaciers may begin surging after a critical mass has built up in the reservoir; information that is accessible with elevation differencing (Kotlyakov et al., 2018; Lovell et al., 2018). Elevation data, and by extension surface slope, can be used to compute and analyse basal shear stress, which may play a critical role in the triggering of surges (Beaud et al., 2022; Thøgersen et al., 2024).

Remote sensing analysis from satellite imagery can produce a large amount of digital elevation models (DEMs), providing observations of the elevation of the glacier surface and its variation over time (e.g., Hugonnet et al., 2021). Such data have been used in numerous studies, ranging from the inventorying of surge-type glaciers to detailed case studies (e.g., Bhambri et al., 2022; Guillet et al., 2022; Guo et al., 2020; Round et al., 2017). However, the use of DEMs for the study of surges is often limited to a few dates or specific case studies, because the temporal availability of DEMs does not always match the surge phases. The retrieval of mass transfer variations happening during such surge events requires dense elevation time series with a resolution of one or a few months in principle. Meanwhile, temporally dense elevation time series from satellites covering a long period of time have recently become available for studying glacier elevation change. Such acquisitions started around the year 2000, with time series now spanning more than two decades, long enough to capture a number of surge events and a few complete surge cycles. In particular, the TERRA satellite with its ASTER sensor is the only optical stereo mission that provides systematic and global acquisitions (Berthier et al., 2023).

Dense elevation time series from this sensor have been successfully used to study long-term elevation trends and multi-year glacier mass balance (e.g., Brun et al., 2017; Hugonnet et al., 2021; Shean et al., 2020). It has also been used multiple times for surge observation with selected DEMs (e.g., King et al., 2021; Zhu et al., 2022; Mattea et al., 2025), and with simple filtering (Lauzon et al., 2023; Li et al., 2023). The DEMs derived from ASTER have an elevation precision of about 5-20 metres and can have large artefacts caused by cloud sensitivity, satellite jitter, or lack of stereo correlation on saturated/textureless terrain (Berthier et al., 2023; Girod et al., 2017). Such noisy DEM time series require specific filtering techniques that preserve surge signals (i.e., preserve elevation observations before, during, and after the surge), as basic thresholds and linear methods used to assess long-term elevation changes might misinterpret surge signals as outliers. Furthermore, the estimate of volume transported and the surface slope are sensitive to data gaps and their interpolation. As a consequence, they need to be computed at similar dates across a whole glacier to ensure physical consistency. Thus, a temporal interpolation of the elevation time series

is required.

60

Various approaches have been implemented in the context of glacier elevation time series analysis. Hugonnet et al. (2021) have implemented a complex workflow for ASTER elevation time series over glaciers at global scale. It captures a number of non-linear elevation changes, but fails to accurately reflect sudden changes associated with surge events. This is due to the filtering and interpolation methods which involve Gaussian Process Regressions, based on a multi-term kernel defined by the variance of elevation changes retrieved at global scale. This method is robust to assess global changes of glacier elevation, but fails to capture the relatively rare surge behaviour. Recent methodological improvements have allowed for sophisticated filtering that are able to preserve abrupt changes in noisy time series of elevation. For example, Wang and Kääb (2015) identified outliers in the absence of a reference elevation using the RANSAC algorithm and Derkacheva et al. (2020) applied a linear non-parametric local regressions (LOWESS) to filter and interpolate non-surge glacier surface velocities. At a higher level of complexity, Shekhar et al. (2021) developed a spline-based approximation framework to model elevation changes with heterogeneous data, which can also be used for filtering. These methodological developments paved the way for processing a large amount of DEMs in a systematic way to study glacier surges from the local to regional scale.

In this study, we develop a workflow to analyse outlier-prone, moderate-precision and high-temporal-resolution elevation datasets adapted to the specificity of surge events. We use established algorithms to filter outliers and interpolate elevations at monthly scale while preserving surge elevation signals. We apply it to an ASTER DEM dataset from Hugonnet et al. (2021) to produce a regional dataset in the Karakoram region covering more than 100 surge-type glaciers. We evaluate the performance of the workflow compared to the results of Hugonnet et al. (2021) and other DEMs (SPOT and HMADEM). We also compare the surge characteristics such as timing and volumes transferred with other studies (e.g., Bhambri et al., 2022; Steiner et al., 2018; Gao et al., 2024).

## 2 Data

In this study, we focus on the Karakoram region (Fig. 1). We use two existing surge-type glacier inventories that cover at least the period 2000 to 2018 in this region (Guillet et al., 2022; Guo et al., 2022). According to Guo et al. (2022), which considers glaciers larger than 0.4 km², there are 354 surge-type glaciers (with individualized tributaries) in the Karakoram and 128 probable or possible ones, representing approximately 8.6% of the regional number of glaciers (39.5% in terms of area). Guillet et al. (2022) identified 223 surge-type glaciers larger than 5 km² not individualizing tributaries. These two studies indicate that surge-type glaciers represent 39% to 45% of the glacierized area in the Karakoram region.

We use the DEMs produced in the global study of Hugonnet et al. (2021) (hereinafter referred to as H21), which ranged from 07/2000 to 09/2019 in the Karakoram and were generated from satellite images of the ASTER sensor. The DEMs have been processed at 30 m resolution with the MMASTER workflow, running under the open-source photogrammetric library MicMac (Girod et al., 2017; Rupnik et al., 2017). All DEMs have been reprojected to 100 m spatial resolution and co-registered to the

**Figure 1.** Map of the study area in the Karakoram, with regional location indicated in the inset map. The colour scale shows the number of pre-processed ASTER-derived elevation observations over the period 2000-2019 from H21. Glacier outlines from RGI7.0 are shown in black (RGI Consortium, 2023). The glaciers with the surge events analysed in section 4 and 5 are outlined in red.

TanDEM-X global DEM (Rizzoli et al., 2017). We use all ASTER elevations estimated by MicMac for any stereo-correlation score, with lower correlation associated with higher uncertainty (H21). Finally, we apply a preprocessing step specific to this dataset: 1) we filter pixels with a difference of more than 400 m between the ASTER DEM and the GLO-90 reference DEM (European Space Agency and Airbus, 2022), 2) we merge the same-date 180 km DEM strips generated by H21 by keeping, in each pixel, the elevation with the highest correlation score. We use the Copernicus DEM GLO-90 as a reference elevation for the coarse filtering of very large outliers. The Copernicus DEM GLO-90 is edited from data of the TanDEM-X mission acquired between 2011 and 2015. The impact of radar penetration in ice and snow (up to about 10 metres) creating a bias in TanDEM-X elevation estimate is negligible compared to the threshold we use (hundreds of metres) (Berthier et al., 2023; Rizzoli et al., 2017).

The sampling is not regular in time and space, and parts of the mountain range have about twice as many DEMs as others (Fig. 1). Overall, 30% (62%, respectively) of the dates in the time series periods are between observations that are less than six months apart (a year, respectively) (Fig. 2, solid orange line).

**Figure 2.** Data gap and temporal coverage of the time series at different processing level. In blue, the proportion of the interpolated on-glacier data gap over the time series period, after the processing workflow. In orange, the proportion of days that fall below the time interval range (e.g., 62% of any date in the time series periods are between pre-processed observations less than a year apart). The x-axis are independent, the y-axis is shared.

#### 105 3 Methods


#### 3.1 Workflow

We present a workflow to filter and interpolate stacks of ASTER DEMs, specifically designed to handle surge events. We use the ASTER DEMs of H21, but processed them with a different workflow, because the H21 workflow performs weakly on surge events (see for example figure S1). We use the H21 workflow as a baseline to compare our own workflow to highlight improvements for surge cases. Our workflow is divided into two main steps (Fig. 3).

First, we filter the dataset to remove remaining outliers in three steps:

- 1. LOWESS workflow, core step of the filtering: we apply an iteration of the LOWESS algorithm (detailed in subsection 3.1.1).
- 2. Morphological 3x3 erosion: we implement a morphological erosion with a 3x3 kernel on the binary data mask. It removes pixels adjacent to outliers, as they also have reduced precision due to the photogrammetric processing.
  - 3. Removal of time series with less than 10 points: we consider such time series not dense enough for our application.

**Figure 3.** Workflow of the elevation time series processing, with an example of time series processed. "it." in the time series legend stands for iteration (of the LOWESS algorithm). The location of the time series exemplified is labelled "TSa" in the caption and map of Fig. 7.c. A version of the filtering of the time series, coloured by the elevation error estimate, is provided in Supplementary Fig. S8.

Second, we interpolate the time series at regular time intervals using a B-spline method which includes an automatic hyperparameterisation algorithm (ALPS-REML), detailed in subsection 3.1.2. The interpolated elevations are provided as a monthly time series.

#### **120 3.1.1 LOWESS filter**

We filter the elevation time series by two iterations of the non-parametric LOWESS algorithm, which is a moving weighted regression (Cleveland and Devlin, 1988; Derkacheva et al., 2020). We use the Python *scikit-misc* implementation. For our dataset, the output of the regression is too sensitive to noise overall and too smooth over surges to be used directly as an interpolation of the elevation, so we use it for filtering only.

Here are the main parameters set for each LOWESS iteration. They have been manually tuned after visual evaluation on a number of time series samples, both with and without surge signal (Fig. 4). We caution that these parameters were chosen specifically for the ASTER DEM dataset and might not all be suitable for other datasets (as discussed in subsection 5.4).

Span: smoothing parameter, expressed as the fraction [0-1] of points of the time series used at each local regression. A
larger value leads to more smoothing. We set it at 0.4 and 0.3 for the first and second iteration, respectively.

- Degree: degree of the local polynomial regression. We choose a degree 2.
- Family: assumed distribution of the errors, with a choice between "gaussian" (fit is performed with a least-squares) and "symmetric" (fit is performed robustly by redescending M-estimators). We use "symmetric".
- Weights: weights to be given to individual observations in the sum of squared residuals. We use the uncertainty provided for each elevation in H21, which models heteroscedasticity (variable error) as a function of slope and the quality of stereo-correlation based on elevation differences on stable terrain.

We use the LOWESS algorithm in the following sequence (Fig. 5): we run two iterations of the LOWESS regression with a decreasing smoothing factor. At each iteration, we compute a threshold envelope around the regression which is used to remove points falling outside of it. The envelopes are derivative-varying to prevent the filter from removing accurate observed signals close to surge events (see example in Fig. 5). We assume fast-varying elevation (high derivative) is a potential surge, and then use a larger threshold. For the first iteration, the threshold is 150 m for fast-varying elevation above 50 m yr<sup>-1</sup> derivative, and then linearly down to 45 m at lower elevation change rate. The threshold is lower for the second iteration: 100 m above 50 m yr<sup>-1</sup> elevation change rate, down to 30 m below. Time series with both large temporal data gaps and a noisy signal can create computational errors for small smoothing parameters. Therefore, at each regression, we implement a step-by-step increase in the smoothing parameter in case of such errors, depicted as the faction value in Fig. 5. In case of computational error remaining after a +0.05 (resp. +0.10) increase of the fraction parameter, we filter out the full time series.

## 3.1.2 ALPS - REML interpolation






Approximation by Localized Penalized Splines (ALPS) is a unified time series modelling framework introduced in Shekhar et al. (2021). ALPS builds on the localized nature of B-spline basis functions to model time series with highly non-uniform sampling. In this research, we use a mixed modelling analogue of the statistical B-spline regression model introduced in Shekhar et al. (2021). This is motivated by the capability of the mixed models to segregate high-frequency and low-frequency components of the overall model, thus allowing us to narrow down the effect of the regularization/smoothing specifically on the high-frequency components. The latter are responsible for the over-fitting behaviour of the model, i.e., the fact that it fits too closely or exactly to the training data and becomes inaccurate for new data. This is particularly problematic for noisy time series like ASTER DEMs.

Another change inherent in our approach, as compared to the approach described in Shekhar et al. (2021), is the model fitting algorithm. The original ALPS model used the Generalized Cross Validation (GCV) metric for estimating the model parameters. However, here we take an alternative route and use the restricted maximum likelihood (REML) approach for fitting our model. The GCV metric quantifies the generalization error of the model by making prediction at data points that were not used to fit the regression model. Hence, the minimization of GCV metric forces the model to predict accurately at unseen locations as described in Wahba (1990). REML on the other hand formulates the problem from a statistical perspective and optimizes the regression parameters so that the probability of observing the data is maximized. A more detailed explanation of REML can be found in Ruppert et al. (2009). The reason for choosing REML over GCV in this work can be attributed to the fact that GCV

**Figure 4.** Impacts of the different LOWESS parameters (rows 1 to 4) and ALPS parameters (rows 5 and 6) on the regression/interpolation solutions. Plain lines are the final selected values. The columns correspond to three different data points (TSa-c, locations shown on Fig. 7.c).

is well known to underestimate model uncertainty, thereby providing over-confident predictions, which in some extreme cases can be misleading. Additionally, for the time series under consideration in this work, the ALPS model with the original GCV based model fitting was over-fitting to noise, making it unsuitable. In order to produce interpolated results in this paper, we use

**Figure 5.** Complete workflow of the LOWESS filter step. The envelopes are the maximum distance threshold allowed between the LOWESS regression and the time series values, which vary with the LOWESS regression derivatives as shown in the inserted plot on the top-left.

the ALPS-REML code provided. After visual tuning of the parameters on a sample set of time series, we set a degree of the basis functions p of 4, and an order of penalty q of 1 (Fig. 4). Note that the confidence interval estimated with the ALPS-REML algorithm and represented on the figures of this article is valid for the interpolation only and not for the whole workflow output.

## 3.1.3 Comparison with Gaussian Process regression

Gaussian Process (GP) regression is a non-parametric method that relies on estimating the data covariance to provide an optimized interpolator (Cressie, 1993; Rasmussen and Williams, 2005; Williams, 2007). Under certain assumptions, including notably second-order stationarity, GP regression has been shown to be the "best linear unbiased predictor". It is the method used by H21 on this same dataset, to compute long-term mass balance estimations worldwide. We use a GP covariance with terms estimated in H21 through a global variogram analysis. This analysis identified several kernel components (periodic, local, linear, etc.), that are not specifically tuned for surges.

We note that, contrary to GP regression, ALPS approximates the data with polynomials under the assumption of a degree of smoothness of the data, with no need for us to inform the behaviour of the data. Although both GP regression and ALPS need domain knowledge to decide the covariance kernel and spline degree/penalty respectively, from a user's perspective using GPs can be more complex owing to the well studied difficulty of optimizing the kernel, mean function and dimensionality (Pu, 2024). For ALPS on the other hand, we simply manually select degrees and penalty orders from a small set of choices.

Reparametrization of the kernel used by H21 gave slightly worse results than those obtained with the ALPS-REML method. Our limitation with GP regression lies in the kernel definition which is done according to the variance of elevation changes. Each surge event is different in variances, which is also very different from the data variance in quiescent periods or on non-surge-type glaciers. We tried different settings of the kernels, that differ from the study of H21. We removed the seasonal term of the model. The length scale and the magnitude parameters of the remaining terms were manually tuned after testing. We added radial basis function terms of length scales of a few months and with a variance of a few tens/hundreds of square meters. The kernels that provided a suitable interpolation were slightly outperformed by the ALPS-REML algorithm. This could be reevaluated for other datasets (for e.g. less noisy), more complex steps or adapted GP regression processes and future advances (e.g., de-trending before GP regression or using other predictors).

## 3.2 Volume transfer estimate







We estimate the volume transferred during surge events by assessing both the positive and negative glacier net volume changes over specific areas. Unless specified, the extent is the surge-affected area manually drawn from the elevation change map calculated over the surge duration. We separate the reservoir and the receiving areas into two distinct polygons. It is difficult to constrain precisely the initiation and termination of surges. The surge dates (Table 1) are estimated visually from two sources: the pre-processed timeseries and the interpolated elevation changes. None of these sources permits us to be sure of the exact month of start or end of the surge. We estimate the dates from interpolated elevation change (e.g. Fig. 8) when computing volume transfers, such "apparent" dates are less exact but capture the overall mass transferred in our generated dataset. We may also estimate the dates from pre-processed time series (not affected by filtering and interpolation defects) for information or validation, which permits us to be more exact although we are still limited by the number of observations. For example, for the time series Fig. S2.a in the Supplement (from Khurdopin glacier), the surge period estimate at this location from the interpolated time series would be around 2016-06 to 2019-02, against 2016-12 to around late-2017 (there is no observation between 2017-06 and 2018-07, thus time series at other locations are required for a better estimate).

To compute the transferred volume, we subtract the interpolated elevation at two dates. We then mask the surrounding areas. We interpolate (small) data gaps in the elevation change maps with a bilinear interpolation. Finally, we retrieve the volume by multiplying the mean elevation change with the delineated area.

The sum of the volume changes in the two areas gives the volume imbalance in cubic metres. We divide the volume imbalance by the surge-affected area to provide the metric imbalance in metres (as if the imbalance was uniformly distributed on the surge-affected area).

## 3.3 Uncertainty of volume transfer estimates

We calculate indicative uncertainties of the volume transfer estimates. These uncertainties do not explicitly take into account possible errors introduced during the filtering and interpolation of each event.

Our uncertainty is estimated with the following formula.

$$\sigma_{\Delta V} = \sqrt{(\sigma h_{\Delta DEM}(p + 5(1 - p))A_{area})^2 + (max(d\Delta V_{-100m}, d\Delta V_{+100m}))^2}$$

The first member of the formula account for the uncertainty in average elevation difference.  $\sigma h_{\Delta DEM}$  is the uncertainty in the mean elevation difference obtained by propagating the pixel-wise measurement uncertainty. The pixel-wise uncertainty is estimated from elevation differences between the interpolated ASTER DEMs and reference DEMs (SPOT5 HRS, SPOT6 and HMA DEM; details in subsection 5.1), considered as the true elevation, over four surge events (Hispar, two dates on Braldu surge, and Kunyang glaciers; Fig. 10) within the surge-affected zone. It is therefore representative of the error on glaciers, during surge events. From each dataset, we reconstruct an empirical variogram using the *SciKit GStat* Python library and all variograms are normalized by their variance and aggregated by taking the mean. We then fit the experimental variogram with a double-range Gaussian model (estimated ranges of 1.4 and 19 km) and estimate the mean elevation difference uncertainty from the number of effective samples calculated from the model with the *xDEM* Python library (Supplementary Fig. S11).  $A_{area}$  is the area of the delineated zone and p the proportion of  $A_{area}$  with valid observations (ranging from 0.92 to 1, median of 0.99). This formulation assumes that the uncertainties of spatially interpolated observations is 5 times larger than the measurement uncertainties, as in Berthier et al. (2014).

The second member of the formula estimates the volume uncertainty due to the manual delineation of the area over which the volume change is computed.  $d\Delta V_{-100m}$  and  $d\Delta V_{+100m}$  are the differences between the volume change estimated over the delineated area and the volume change estimated over an area with a buffer of -100 or +100 m, respectively. This assumes an uncertainty in our manual delineation of 1 pixel, which is reasonable given the strong contrast in elevation on the edges of the surge reservoir and receiving areas.

We propagate the uncertainties to the volume imbalance, assuming independent errors, with the following equation:

$$\sigma_{V\ bal} = \sqrt{(\sigma_{\Delta V\ reservoir})^2 + (\sigma_{\Delta V\ receiving})^2}$$

The uncertainty in metric imbalance is then expressed as  $\sigma_{V\_surface\_bal} = \frac{\sigma_{V\_bal}}{A\ total}$  with  $A\_total$  the total area considered.

## 4 Results






## 4.1 Performance of the outlier filtering

We compare the filter and the temporal interpolation developed in this study with those of H21 in locations that are affected by surges, but also for all glaciers in the region (Fig. 6, Fig. 7). In H21, the iterative GP regression filtering is responsible for removing some high-amplitude surge signals (Fig. 6.c1-2, or abnormal gap A1 circled in red in Fig. 7.a). In H21, the kernel of the GP regression filter does not model well the elevation change that is typically observed during some of the surge events

(e.g., Fig. 6.c1). In our workflow, the LOWESS filter behaves with varying performance, depending on the time series quality (noise, temporal density, surge amplitude). It preserves well the surge signal of 3 of the 4 events we analyse in subsection 4.3, and this observation seems to extend to a number of surge events in the Karakoram. One exception is periods of low temporal density during surge events, especially when combined to strong melt before and after the surge. A typical example of such erroneous filtering is a part of the front of Khurdopin glacier (Supplementary Fig. S2.a). In this time series, two critical observations are filtered out around 2017 during the short surge. The ALPS-REML interpolation smooths the signal even further, as both LOWESS and ALPS fits are sensitive to the lack of elevation measurements at abrupt trend changes. Strong melt in the receiving area increases the elevation-change smoothing effect of the fits by reducing the average elevation change locally before and after the surge.

The LOWESS workflow is also sensitive to the weight estimate and noise in textureless and steep areas, for example, resulting in the filtering being oversensitive to noise compared to the original workflow (red circles B1-2 in Fig. 7.b). This filter oversensitivity occurs on time series with scattered elevations, and it is often due to the correlation score that is not very representative of the actual pixel quality: outliers may have lower uncertainties than more accurate observations (e.g., Supplementary Fig. S2.e or S7 at 15 km). These types of location are not predominant in surge-affected areas, and a number of them are completely filtered out during subsequent filtering steps. Thus, filtered areas (data gaps) and spurious elevations are more prevalent with our method than with the filter of H21 over textureless accumulation areas.

In summary, our filter better preserves the surge signals that were filtered out in the workflow of H21. However, the new filter is more noise-sensitive over textureless accumulation areas and rough terrain, leading to data gaps or artifacts with large elevation changes. The preprocessing step removed 46% of the original regional dataset (number of on-glacier pixel), and the filtering step removed a further 42% of the preprocessed dataset (69% removed in total compared to the original dataset). After filtering, nearly 30% (62%, respectively) of any date in the time series periods are between observations less than 9 months apart (one and a half years, respectively). Before filtering, for the same percentage, it was a half-year (one year, respectively) (Fig. 2, solid orange line). The time series are about half as dense as before, temporally.

#### 4.2 Performance of the temporal interpolation




The interpolation of H21 is a GP regression with the same kernel as for the filtering. Fig. 6.a-b1 shows edge effects at the temporal bound of the time series due to the linear term of the kernel. It is noteworthy to mention that by its design, the original kernel is optimized to preserve a linear trend to extrapolate out of the observation period of each pixel. The seasonal term of the kernel creates the one-year periodicity. In comparison, our workflow shows only limited border effects. The workflow presented in this study better fits changes in trends (ex. Fig 6.a1-2), and preserves most of the surge signal (Fig 6.c2). However, dense clusters of points are regularly over-fitted, creating spurious high frequency oscillations spanning typically about 6 to 12 months, as illustrated in Fig. 6.c2 around 2006 and 2011 or in Fig. 6.a2 around 2006. Comparing the final interpolated elevation changes over two years (Fig. 7.c-d), our workflow can capture the complete surge signal of Hispar and Braldu glaciers (red circles C1-3 in Fig. 7.c), which was not the case for the previous workflow. At these locations, the original method of H21 completely filters out the surge signal, filling the period with the global trend or a completely smoothed trend (e.g.,

**Figure 6.** Comparison of the filter and interpolation methods: (a1-c1) from H21 against (a2-c2) the workflow presented in this study. The three time series all show a surge around 2015. Their location is represented on the map Fig. 7.c (points TSa-c). We avoid overlaying points for readability (i.e., points exist but are masked in lower-level time series, in legend order). The confidence interval is valid for the interpolation only and not the whole workflow output: it is the 1  $\sigma$  standard deviation credible interval for GP regression (H21), and it is the 95% confidence interval for ALPS-REML (Shekhar et al., 2021)

Supplementary Fig. S1). Moreover, several reservoir or receiving areas of the surges show smaller elevation changes with the original method, which tend to smooth remaining surge signals, both in time and in elevation (e.g., Fig. 6.c1 and Supplementary Fig. S2.d). The maximum spatial coverage of on-glacier interpolated elevation over the Karakoram is around 80% from 2005 to 2015 (Fig. 2, solid blue line).

**Figure 7.** a-b: Maps of maximum elevation change after filtering. c-d: Elevation change maps over two years (Hispar glacier surge period). The green points and their labels (TSa-c) in c) correspond to the localisation of the time series in Fig. 6 (a-c). Their coordinates are (EPSG:4326): TSa (75.863, 36.055), TSb (75.295,36.089) and TSc (75.861,36.200). The green lines on d are the centrelines of the studied glaciers. The red circles (A1-C3) and the dotted lines (a2-4 and d3) show or delimitate areas discussed in the text. The insets for Kyagar glacier have the same scale as the main frames.

## 4.3 Analysis of selected surge events


To illustrate the outputs of our method, we analyse four surge events that have been studied in the literature. They occur on four glaciers: Hispar, Khurdopin, Kyagar and Yazghil glaciers. Fig. 8 shows the spatio-temporal evolution of the glaciers surface

elevation along their centreline (green line on Fig. 7.d). Time series, extracted at regular intervals along the selected centrelines of each glacier are shown in Supplement (Fig. S3 to S7), and surge volume transfers are reported in Table 1 for each glacier.

## 4.3.1 Hispar glacier






We observe the influence of Kunyang tributary surge that reached Hispar main glacier tongue (around kilometre 40) in early 2008 (Fig. 8.a., area a1). The surge front propagates downstream for several years with a decreasing propagation rate (2009-2012; Fig. 8.a, area a1), while strong thinning starts at the junction and approximately 5 kilometres upstream of the surge front. Meanwhile, a slight and more regular build-up or thickening occurs above, upslope of 25 km (Fig. 8.a, area a6). The surge of Hispar main trunk seems to start in early to mid-2014 and end around June 2016 (area between the lines a2 and a3 on Fig. 7.d and Fig. 8), with small mass displacement until the end of 2017, downslope of the Kunyang junction. Sharp spurious highfrequency oscillations of positive and negative elevation changes from mid-2013 to mid-2014, which we attribute to artefacts of our method, are visible horizontally on Fig. 8.a. The time series shows dense and very scattered elevation observations at this period even on stable ground (Supplementary Fig. S2.c), causing these artefacts. This spread may be due to tilts or undulations remaining in the DEMs. The results indicate that the dynamic balance line location is not stable in time. On the branch of Hispar Pass (head of one of the main branches, location on Fig. 7.d), the reservoir area extends from 5 km from the pass, at an icefall (line a2 on Fig. 7.d and Fig. 8), down to 20 km from the pass at the junction with the Yutmaru tributary in the first part of the surge. From the end of 2015 to the termination of the surge, the reservoir area limit propagates down by 5-10 kilometres (below the junction) (line a5 on Fig. 7.d and Fig. 8). We plot an elevation time series at this location (Fig. 6.b2, location TSb on Fig. 7.c). The receiving area extends from the end of the reservoir area at 20-25 km from the pass along the centreline, down to nearly 40 km from the pass at the junction with the Kunyang tributary (line a3 on Fig. 7.d and Fig. 8.a).

## 4.3.2 Khurdopin glacier

Khurdopin glacier has a strong mid-glacier thickening signal until the surge onset. The distinct area of positive elevation-change trend extends down-glacier during at least 15 years (Fig. 8.b, area b1). This mass build-up may be the geometry readjustment of the glacier in its quiescent phase, after the previous surge in 1998 (Quincey et al., 2011). The lower limit of this build-up area propagates downward from about 25.5 km of the glacier head in 2001 to about 33.5 km in 2015. The limit advances approximately 600 m per year during this period, which is about 7 times faster than the surface velocity (measured 2 km upstream of the front), according to velocities (temporal baseline from 300 to 430 days) from the NASA MEaSURES ITS\_LIVE project repository (Gardner et al., 2022). During this period, we do not observe a clear mass transfer from an upper reservoir area, which thus seems different from a slow surge onset. The upper limit of the build-up area (which will mostly become the reservoir area) is stable in time, at the bottom of two icefalls for the two main branches just above their junction.

The surge starts in 2016, with the build-up front becoming a surge front with a higher propagation rate. Both our filter and interpolation methods here fail to fully capture the surge signal of the receiving area (see discussion section 5.3). This failure

leads to an apparent surge end in early-2019 on interpolated data, which is overestimated by about a year and a half according to non-interpolated time series (Supplementary Fig. S2.a). A distinct and local positive elevation change pattern is visible after the surge around kilometre 23 (Fig. 8.b, area b2).

## 4.3.3 Kyagar glacier



Kyagar glacier is located about 110 km east of the other glaciers (Fig. 1). A slight mass build-up is visible since the beginning of the time series in the first 10 kilometres of the glacier, and extends down to about 14 km a few years before the surge (Fig. 8.c, area c1). The surge as visible on interpolated data starts in 2013 or the beginning of 2014, and ends around 2016 (Fig. 8.c, area c2). However, the actual surge is certainly shorter. The beginning of the surge appears sooner in the interpolated time series, and the end is also represented nearly a year later from what is visible on the non-interpolated time series of most of the receiving area. During the surge period, there are about 1-2 observations per year. An area or poor quality in the ASTER time series results in artefacts after processing, at 5 km from the glacier head, which is located around the equilibrium line of the glacier (Fig. 8.c, area c3). This area seems to be in the reservoir area, therefore causing a bias in the volume transfer calculation. We manually draw a mask to remove artefacts for a better estimate (Table 1).

## 4.3.4 Yazghil glacier

Our dataset captures a full surge cycle of Yazghil glacier. On this glacier, the surge signal has a low amplitude (approximately ten metres) compared to the time series, and thus noise is often overfitted resulting in frequent interpolation artefacts. Some seasonal signal seems also to be fitted, for example during the period 2013-2016 thanks to denser and consistent time series (horizontal lines on Fig. 8.d). A surge starts around August to November 2003 and ends around October 2006 to February 2007 (Fig. 8.d, area d1), and a new surge starts in 2016 or 2017 (the end is not captured; 8.d area d2). The build-up phase of the second surge is visible, representing about half of the quiescence phase (Fig. 8.d area d3, delimited by dotted lines d3 on Fig. 7.d). One of the tributaries of Yazghil glacier (junction at km 18) is also surge-type, and seems to have surged during our study period in about 2008-2013.

**Figure 8.** a-d: Interpolated surface elevation time series along the centreline of four glaciers (in green in Fig. 7.d). Glaciers flow from left to right on the different panels. Note that the colorscales represent different elevation change rate amplitude and that they are non linear.

| Clasian        | Data ataut    | Doto and      | Reservoir                                  | Receiving                                |                                            |
|----------------|---------------|---------------|--------------------------------------------|------------------------------------------|--------------------------------------------|
| Glacier        | Date start    | Date end      | vol. change                                | vol. change                              | Imbalance                                  |
| RGI 7.0 code   | [time series] | [time series] | [Surface area]                             | [Surface area]                           |                                            |
| Hispar         | 2014-01       | 2016-09       | $-2421 \pm 374 \times 10^6 \text{ m}^3$    | $3108 \pm 177 \times 10^6 \text{ m}^3$   | $687 \pm 414 \times 10^6 \text{ m}^3$      |
| 21670          | [2014-05]     | [2016-06]     | $[106  \mathrm{km}^2]$                     | [48 km <sup>2</sup> ]                    | $4.46 \pm 2.69 \text{ m}$                  |
| Yazghil        | 2003-07       | 2007-01       | $-32 \pm 30 \text{ x } 10^6 \text{ m}^3$   | $63 \pm 26 \text{ x } 10^6 \text{ m}^3$  | $32 \pm 40 \text{ x } 10^6 \text{ m}^3$    |
| 21865          | [2004-01]     | [2006-08]     | [8 km <sup>2</sup> ]                       | [6 km <sup>2</sup> ]                     | $2.20 \pm 2.77 \text{ m}$                  |
| Khurdopin      | 2016-03       | 2019-03       | $-813 \pm 136 \text{ x } 10^6 \text{ m}^3$ | $713 \pm 64 \times 10^6 \text{ m}^3$     | $-100 \pm 150 \text{ x } 10^6 \text{ m}^3$ |
| 14958          | [2016-04]     | [2017-07]     | $[33 \text{ km}^2]$                        | [15 km <sup>2</sup> ]                    | $-1.9 \pm 1.64 \ \mathrm{m}$               |
| Kyagar         | 2012-11       | 2017-01       | $-271 \pm 92 \times 10^6 \text{ m}^3$      | $269 \pm 55 \text{ x } 10^6 \text{ m}^3$ | $-2 \pm 107 \text{ x } 10^6 \text{ m}^3$   |
| 14958          | [2013-10]     | [2015-12]     | [21 km <sup>2</sup> ]                      | [8 km <sup>2</sup> ]                     | $-0.07 \pm 3.64 \text{ m}$                 |
| Kyagar without | _             | _             | $-217 \pm 116 \text{ x } 10^6 \text{ m}^3$ | $269 \pm 55 \text{ x } 10^6 \text{ m}^3$ | $52 \pm 128 \text{ x } 10^6 \text{ m}^3$   |
| artefact       |               |               | $[20 \text{ km}^2]$                        | [8 km <sup>2</sup> ]                     | $1.33 \pm 3.54 \text{ m}$                  |

**Table 1.** Timing and transferred volume of the surges of four glaciers in the study area. The main dates are given according to the interpolated elevation time series on the centrelines (Figure 8). We compute the transferred volume ("vol. change") from interpolated DEMs at these dates to estimate the corresponding volume change from both reservoir and receiving areas. The dates between brackets are those estimated visually on non-interpolated time series, thus less smoothed, given for indication. They are not accurate to the month due to ASTER acquisition dates. The volume change and the imbalance computation method is detailed in subsection 3.2. For these glaciers, the percentage of data gap after the workflow presented in this study is ranging from 0 to 5.6% (median 1.4%), and after bilinear interpolation it is 0 to 0.8% (median 0.2%). The prefix of RGI codes is "RGI2000-v7.0-G-14-" (RGI Consortium, 2023).

## 5 Discussion

## 5.1 Processing quality

To assess the quality of our results, we 1) compare our interpolated elevations with external DEMs produced from high resolution satellite imagery, and 2) test the sensitivity of the interpolation to data gaps.

We compare the interpolated elevation with external DEMs, produced from optical very-high resolution satellite imagery (Fig. 9). This comparison provides a validation of estimated elevation during a few surge events. We use SPOT5 HRS and SPOT6 DEMs generated by Berthier and Brun (2019), and along-track HMA DEMs (Shean, 2017) (list in Table S2 of the Supplement). We co-register each external DEM on the ASTER interpolation on stable terrain. The Normalized Median Absolute Deviation (NMAD) after co-registration ranges from 6.8 to 15.6 m (median 7.4 m), which shows good agreement with discrepancies of a few meters. Extreme cases occur locally, with differences reaching tens of meters, but it is generally unclear which dataset is flawed. The case study of Khurdopin glacier surge shows that a wrong estimate of a hundred meters of

our workflow can occur on exceptional events and at precise dates during the surge (Supplementary Fig. S2.a). The map of elevation differences on the glaciers shows differences of a few meters overall, which is moderate compared to the amplitude of the surge elevation change (Fig. 10). The difference may be important, such as several tens of metres locally at the surge front (e.g., Fig. 10.a-b at the Kunyang-Hispar junction). Across the entire glacier areas, consistent discrepancies are observed. For instance, on 2015-10-13, Hispar glacier exhibited a median difference of -4.3 m with a standard deviation of 9.7 m. Similarly, on 2015-11-28, Braldu glacier exhibited a median difference of -5.2 m with a standard deviation of 8.7 m. Larger local differences are located around the surge front: e.g., up to 24 m at Hispar surge front on 2015-10-13. The elevation difference values during a surge event and during quiescence do not show important differences at the scale of the surge-affected area (Fig. 11). The discrepancy associated with a surge period is overall of the same magnitude as other noise, considering the large dispersions.

360

365

350

355

One of the main limitations of our results comes from the relative temporal sparsity of the input observations. Here, we investigate the impact of data gaps on our interpolated time series. Some parts of our study area are characterised by a low temporal density of observations during surge events (e.g., less than three observations per year) (Fig. 1). In such situations, our method of filtering and interpolation usually leads to an underestimate of the transferred volume and an overestimate of the surge duration (e.g., twice its duration for Kyagar glacier). Onset and end dates cannot be determined precisely between two observations separated by more than 6 months or a year, even on filtered series, as it occurs for Kyagar glacier.

To test the sensitivity of the ALPS-REML method to data gaps, we interpolate an elevation time series after removing all points in a 450-day moving window (Fig. 12). Each iteration results in a period of at least 450 days without observation, which is common in the filtered series. For instance, on the surge-affected area of Kyagar glacier, which is subject to a lack of observation for our processing, there are on average three time intervals of 400-to-500 days without observations per time series (one time interval for Hispar glacier, in comparison). For the selected time series Fig. 12.a and Fig. 12.c, the test shows strong smoothing, although the surge signal is still visible over large time frames. The interpolated dates of the surge onset (respectively ending) are advanced (respectively delayed) by up to two years compared to the original interpolation. The surge elevation change can be underestimated by up to 20 meters. This can be larger for longer time gaps or surges with stronger elevation changes before or after the surge. The case shown on Fig. 12.b is specific, as it lies close to the dynamic balance line (in the receiving area at an early stage of the surge, and then in the reservoir area). The surge signal is completely smoothed out when data gaps occur in the middle of the surge. Other specific surge cases, with limited elevation changes but with strong melt or strong build-ups before or after the surge, could be prone to the same problem.

## 5.2 Comparison with elevation change from H21

We assess pixel level differences in elevation change estimate between the processing workflow of H21 and this workflow. Previous figures showed local differences; here we compare the elevation changes of pixels belonging to eight surge events (Fig. 13, individual graphs on Supplementary Fig. S9). There is a strong smoothing of the original dataset which tends to filter the positive elevation changes occurring in surge receiving areas. They are better interpolated by our workflow (Fig. 13 zone A).

**Figure 9.** a-c: Comparison between elevations from SPOT DEMs (SPOT5 HRS and SPOT6) and HMA DEMs and ASTER elevations interpolated at the same dates. The time series are identical to previous ones (TSa-c in the panel order, Fig. 7.c). The confidence interval is valid for the interpolation only and not the whole workflow output.

No symmetric pattern is visible for negative changes in reservoir areas, probably due to the smaller rates of elevation changes. This erroneous filtering occurs mainly for surges with important and rapid elevation changes: surges of Hispar, Braldu, and Kunyang glaciers (Supplementary Fig. S9), and to a lesser extent of Khurdopin glacier surge. For such glaciers, major differences in total volume change are expected. This is clear in the transferred volume estimates from the original dataset of H21 on Hispar and Khurdopin glacier surges (Supplementary Table S1). Other glaciers also have smaller estimated volumes than with our method, but with smaller discrepancies. Compared with H21, our method finds larger absolute rates of elevation changes (pattern B on Figure 13), probably due to the stronger smoothing of H21 (e.g., Fig. 6.a1 or Supplementary Fig. S2.d). On the other hand, our method creates some artefacts, especially in the accumulation areas where elevation changes are close to zero (zone C on figure 13). This is the case for Kyagar and Braldu glacier surges (Supplementary Fig. S9).




This figure also illustrates the unequal distribution of elevation changes between the reservoir and receiving areas, which is observed for all analysed surges (Fig. 13). Elevation changes are consistently much larger in the receiving areas, whether the glacier front is advancing or not. This is balanced by the extent of the reservoir areas which are larger than those of the receiving

**Figure 10.** a-d: Elevation difference between SPOT DEMs (SPOT5 HRS and SPOT6) and HMA DEMs against ASTER DEMs interpolated at the same dates. The areas selected are Hispar glacier (a, surge in 2014-2016), its Kunyang tributary (b, surge in 2007-2008), and two over Braldu glacier (c-d, surge in 2013-2016). The panels have the same colour range. The green dots show sampled time series (Fig. 6, 7.c and 9).

areas.



On a larger scale, we compare the individual glacier average elevation change between H21 and this workflow for the period 2005-2015 (Supplementary Figure S10). The mean elevation changes are more negative with our workflow (by about 0.44 m for the median value). The discrepancy is larger for surge-type compared to non surge-type glaciers (0.57 and 0.31 m with standard deviations of 1.1 and 1.02 m, respectively). Considering the better retrieval of positive elevation changes of our workflow for surges, we would expect a positive discrepancy for surge-type glaciers. A number of glaciers have artefacts in our dataset, especially negative elevation changes in accumulation areas. At regional scale and possibly glacier scale, the impact of noise may exceed the impact of the improved estimate in areas of positive changes, due to the small number of surge events happening during this period. For calculating geodetic glacier mass balance, the H21 dataset is therefore the preferred choice for non-surge-type glaciers or quiescent periods, and a validation of the elevation interpolated by our method is recommended.

**Figure 11.** Histograms of the elevation difference between the reference DEMs and the DEMs of our workflow interpolated at the same dates. We consider only surge-affected areas. Vertical dotted lines are the median of each histogram. The largest median is 5.18 m (resp. -5.63 m) during surge (resp. during quiescence).

## 5.3 Comparison of surge characteristics with the literature

# 5.3.1 Hispar glacier




Regarding the surge of the main trunk of Hispar described in section 4.3, our date estimates from both interpolated and preprocessed time series (early-2014 to mid-2016) are close to the date estimated in previous studies (autumn 2014 to mid-2016), which were based on remotely sensed velocities (Guo et al., 2020; Paul et al., 2017). Paul et al. (2017) notice a 6-month stop of the surge front around 35 km, up to mid-2015 which is slightly visible here at a similar time (Fig. 8.a, line a3). The fact that the reservoir area does not extend above the icefall has already been observed on other glaciers, including Khurdopin in our study (Nolan et al., 2021; Echelmeyer et al., 1987). The displacement of the dynamic balance line during this surge has not been mentioned in other studies for Hispar, as the data they use (velocities and a limited number of DEMs spaced in time) may not allow this phenomena to be observed (Guo et al., 2020; Paul et al., 2017; Rashid et al., 2018). However, the phenomenon has already been reported and attributed to variations in driving stress (Burgess et al., 2012). Bhambri et al. (2022) estimate volume changes over the period 2014-2020 from ASTER DEMs of -2785 x $10^6$  m<sup>3</sup> in the reservoir area, and 258 $1.6 \pm 465$  $x10^6$  m<sup>3</sup> in the receiving area. Our estimate for the reservoir area differ by 13%, and 20% in the receiving area (Table 1). The smaller volume estimated by Bhambri et al. (2022) may be explained by the melting of the deposited ice volume during the three or four years that separate the surge termination and elevation observations. If we extend the period of volume change calculation from 2014-10 to 2018-08 (the latest date before large data gaps in our time series) to better match that of Bhambri et al. (2022), we estimate a volume change of  $-2255 \pm 181/2634 \pm 410 \times 10^6 \text{ m}^3$  (19% and 2% difference, respectively) closer to their estimate. The differences are within uncertainties, although there is a two years difference between the two estimates

**Figure 12.** Sensitivity of our interpolation method to large data gaps. For the three selected time series (TSa-c of Fig. 6 and location visible on Fig. 7.c), we remove points during 450 continuous days over a moving window and run the interpolation, displayed with orange lines.

#### 425 periods.



The difference between our estimated volume gain and loss is equivalent to a layer of  $4.46 \pm 2.69$  m thickness over the surge area. This imbalance is unexpected as the surge occurs over a short time period and mass should be roughly conserved. The imbalance is quite similar when using two filtered ASTER DEMs over a similar period, instead of the interpolated series, or when calculated over the full glacier system instead of over the delineated reservoir/receiving areas. Another possible source of imbalance is the impact of crevasse opening during the surge, which can represent a non-negligible volume change. As an example, the opening of crevasses can be equivalent to up to 0.2 m thickness at regional scale of the Greenland Ice Sheet (Chudley et al., 2025). As inland parts of these regions are largely crevasse-free, we can expect such impact on the volume to be significantly larger over the highly crevassed post-surge surface of Hispar glacier. By mid-2018 our imbalance is close to zero, as well as is the imbalance of Bhambri et al. (2022) with an end term in 2020, when a number of crevasses have already closed. Khurdopin and Kyagar glaciers were already highly crevassed before the surge, and such crevasse opening effect may be less important.

**Figure 13.** Histogram of interpolated elevation change comparison over 8 surges between the original processing from H21 and this workflow. The superimposed histograms of the 8 surge events are represented individually in Fig. S9 of the Supplement. The elevation changes are retrieved over the surge-affected areas and the surge period estimated from the interpolated elevation time series on the centrelines. The areas and trends designated in red are discussed in subsection 5.2. They highlight areas of large surge smoothing or removal (zone A) or overall smoothing of elevation changes (trend B) by the original method (H21), and artefacts created by the presented workflow (zone C).

#### 5.3.2 Khurdopin glacier




We now discuss the recent surge of Khurdopin glacier (March 2016 to March 2019; Table 1). The geometry readjustment and the propagation of a build-up front during quiescence have not been described on this glacier, to our knowledge. The existence of kinematic waves or surge fronts that propagate the surge instability have regularly been observed on other surges (e.g., Cuffey and Paterson, 2010; Kotlyakov et al., 2018; Turrin et al., 2013), with unclear definition of the phenomena. For Khurdopin glacier, the mechanism seems different from both a kinematic wave or a slow surge onset. As opposed to these processes, here we observe a constant thickening after the downward extension of the build-up area with no upper reservoir area drained. Turrin et al. (2013) observed, with velocity data, the propagation of a surge front (moving as a kinematic wave) several years before the surge of Bering glacier, triggered by the passing of the front through the reservoir area. The build-up lower limit for Khurdopin also propagated faster than the surface velocity. The surge started in October 2016 according to Imran and Ahmad (2021), about 7 months later than our estimate (Table 1), and late August 2015 according to Steiner et al. (2018). The volume received in the receiving area is estimated at  $1182 \times 10^6 \text{ m}^3$  during late August 2015 (elevation extrapolated linearly from TanDEM-X in 2011) to May 2017 (ASTER) data (Jakob Steiner, personal communication). Our estimate over a similar period (2015-09-01 to 2017-06-01) is  $426 \pm 34 \times 10^6 \text{ m}^3$ . Both estimates do not agree, although we do not have an uncertainty

estimate for one volume. Our filter and interpolation methods fail to fully capture the surge signal of the receiving area, in the lower part of the glacier (Fig. 8.b area b3). This failure is due to a low point density combined with a strong thinning signal after the surge (Supplementary Fig. S2.a, in 2017). The filtering workflow removes some of the 2-3 DEM acquisitions over 455 2017 and 2018, which have credible values, May 2017 is the month with the largest difference between the DEM observations and the interpolation, with an elevation change underestimation that reaches 100 m compared to the pre-processed time series. Over a portion of the receiving area, the apparent surge signal duration after interpolation is about 3 years instead of approximately 1 year on pre-processed time series, and may miss locally a maximum of 40 m (about 30%) of the surge elevation amplitude over these three years. Our estimate of the transferred volume in Table 1 is thus underestimated in the receiving area. 460 Our uncertainty estimate is also largely underestimated, as it does not take into account the erroneous filtering. The difference of the pre-processed DEMs from 2015-08-20 and 2017-05-21 shows a cumulative positive volume change of 650 x10<sup>6</sup> m<sup>3</sup>. It is 153% more than with the interpolation, yet nearly half of the estimate of Steiner et al. (2018) which may be also partially overestimated due to their linear extrapolation, as the 2000-2011 trend does not accounts for the later build-up front propagation that we observe. The maximum thickness gain noted by Steiner et al. (2018) was 160 m over this period, against 122 465 m with our pre-processed DEMs (70 m on interpolated DEMs). The case of Khurdopin surge shows that our workflow may be inefficient to preserve a surge signal, in the case of a low number of observations, worsened by strong thinning outside the surge period.

# 5.3.3 Kyagar glacier

Kyagar glacier is located in an area of poor ASTER coverage, compared to other selected glaciers (Fig. 1). During the surge period, there are about 1-2 observations per year, which leads to a smoothing of the surge signal during interpolation. Thus, the onset and ending are visible around end-2012 and early-2017 on interpolated data, while pre-processed time series lead to a more restricted estimate of mid or end-2013 (sooner observation in October after a 14-month data gap) to December 2015. Round et al. (2017) uses satellite imagery to compute velocities and precisely describe the surge development. They find a surge onset in May 2014 after a pre-surge acceleration of 2.5 years, and a surge end between July and August 2015 with limited deceleration later. Li et al. (2023) find very similar timings, plus a continuing deceleration in 2016-2019. Gao et al. (2024) report similar timing, although considering a re-acceleration in 2016 as part of the surge. Gao et al. (2024) estimated the volume transported from ASTER DEMs. During July 2012 to December 2017, they estimate the received volume to be 321 ± 12 x10<sup>6</sup> m³, compared to 262 ± 46 x10<sup>6</sup> m³ with our interpolated data. Their reservoir area volume change estimate is -383 ± 480 30 x10<sup>6</sup> m³, against -326 ± 96 x10<sup>6</sup> m³ for our dataset over the same dates and approximative area (-283 ± 104 x10<sup>6</sup> m³ with bilinear interpolation of the area affected by artefacts). It represents differences in transferred volume estimate of 18% and 15%.

# 5.3.4 Yazghil glacier

Yazghil glacier has not been extensively studied. Bhambri et al. (2017) date the last surge in 2006, with a gradual increase in velocities before this year. The study estimates from 1972-2016 data that Yazghil glacier has a cycle length (surge repetition period, including quiescence and surge durations) of about 8 years, among the shortest surge cycles in HMA (Bhambri et al., 2017; Sun et al., 2022; Vale et al., 2021; Yao et al., 2023). The next surge, which was expected to occur around 2014 based on the cycle length, had not started by the end of 2016, according to the study. Our data suggest that it started 1-2 years later, implying a quiescence phase of 11-13 years for this cycle.

490

495

## 5.3.5 Conclusion of the case studies

Overall, the dataset produced by our workflow compares well with existing observations from the literature. The surge dates and the estimated transferred volume agree, except for the date of Kyagar surge and the transferred volume of Khurdopin surge (Table 1). The order of magnitude of the imbalances corresponds to the order of magnitude of the measurement uncertainty. For the two critical cases (Kyagar and Khurdopin surges), the workflow shows its limitations in the case of a low number of DEMs, worsened in the case of a strong thinning signal outside the surge period (Khurdopin surge). Our dataset offers new insights on some undescribed processes in these studies, such as the displacement of the dynamic balance line of Hispar surge or the propagation of a surge front during the build-up phase preceding Khurdopin surge.

## 5.4 Applicability to other datasets

Here we discuss the feasibility of applying the proposed workflow to different datasets, possibly including several data sources to increase temporal resolution (i.e., DEMs from different sensors). Even in the case of a similar ASTER DEM dataset processed differently, with lower noise/higher precision, several changes may be made to adapt the filtering. For denser series, a diminution of the span parameter along with a diminution of the filter threshold in the LOWESS workflow should be tested. Abandoning morphological erosion should also be considered. It addresses an issue specific to the photogrammetric processing which tend to affect pixels neighbouring outliers. Deleting this step would be beneficial given the large number of pixels it removes. The use of weighting could also be abandoned in the case of more precise DEMs, as the uncertainty values are not completely representative of the confidence in the measurement. The ALPS-REML prediction parameters could remain unchanged, although the hyperparameters degree of the basis functions *p* and the order of penalty *q* can be modified to adjust the smoothing and border effects. More complex considerations would be required in the case of several data sources. More particularly, the weighting may be defined differently to ensure consistency between the datasets.

## 6 Conclusions

520

525

530

We present a new workflow for processing DEM time series of high temporal resolution that is specifically designed to preserve the elevation signal of glacier surge events. We applied the workflow to a dataset from the ASTER sensor over 2000-2019. We filter the data with a LOWESS algorithm, which preserves the surge signal. Some filter issues can appear in difficult areas, which are often not located in surge-affected areas (e.g. textureless accumulation areas, steep slopes). The elevation interpolation (B-spline method ALPS-REML) allows for the observation of surge dynamics, and the estimate of mass transfers at a monthly interval. Surge events with too few DEM observations tend to be smoothed, resulting in an underestimation of the surface elevation change and surge duration. In our study area in the Karakoram range (HMA), our method provides interpolated time series for 80% of glacier pixels. Our workflow better preserves surge events compared to the original non surge-specific workflow. The data obtained are fairly comparable to those from independent studies on several events, except in a few cases. We find discrepancies in the estimated transferred volumes compared to previous studies ranging from 2% to 19% on two surge events and four volumes transferred, and 64% on Khurdopin surge. The workflow, applied to ASTER DEMs but which can be adapted to other datasets, can generate a unique elevation time series able to represent thickness changes of surge events on a monthly scale over a regional extent. It opens new possibilities for the combined analysis of elevation and velocity change during surge events, or more complex derivatives such as surface slope and driving stress.

Code and data availability. Although the study of Shekhar et al. (2021) only describes the ALPS-GCV implementation, the code provided with that study in the repository Shekhar (2020) also contains the implementation of ALPS-REML, which was used without changes in our study except for parameters declared in the section 3.1.1. The code of our workflow can be found at the following repository: https://doi.org/10.5281/zenodo.14045604 (Beraud et al., 2024). Sample data of elevation change and surge-affected areas for the four selected glaciers are also available in that repository. Finally, data including interpolated datasets covering Karakoram and other mountain ranges affected by surge events in HMA will be added in future versions of the reposity, during the months following publication.

Author contributions. Luc Beraud: Conceptualization, Methodology, Software, Writing - Original Draft. Fanny Brun and Amaury Dehecq: Conceptualization, Methodology, Writing - Review & Editing, Supervision. Romain Hugonnet and Prashant Shekhar: Software, Writing - Review & Editing.

535 Competing interests. The authors declare no competing interest.

Acknowledgements. We are grateful to Laurane Charrier for her suggestions and methodological input, and to Adrien Gilbert for our discussions about surge case studies. We thank Etienne Berthier for providing SPOT DEMs and discussing the methodology of the work. We also

| thank Jakob Steiner for discussions and additional details related to his published work. We finally thank the French program "Programme National de Télédétection Spatiale (PNTS)" for its funding support. |  |  |  |  |  |  |
|--------------------------------------------------------------------------------------------------------------------------------------------------------------------------------------------------------------|--|--|--|--|--|--|
|                                                                                                                                                                                                              |  |  |  |  |  |  |
|                                                                                                                                                                                                              |  |  |  |  |  |  |

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
