# Peer review of "Glacier surge monitoring from temporally dense elevation time series: application to an ASTER dataset over the Karakoram region"

_EGUsphere, 2024_

## Referee Comment (RC3)

**Review of 2024-3480**

The manuscript proposed by Beraud et al. presents a new methodology to filter outliers and estimate glacier surface elevation from dense digital elevation model time series.
Their study relies on the same data as the Hugonnet et al (2021) study.
Instead of Gaussian Process Regression, Beraud et al implement an additional LOWESS filtering step, before feeding the fitlered time series to a localized b-splines scheme in order to interpolate surface elevation measurements.

While the work and the methods shows promise, I strongly recommend that the authors take the necessary steps to strengthen the manuscript.
The paper appears to be in an early stage of development and would benefit from significant revisions before it is considered for publication: sections 4 and 5 are particularly challenging to understand. With the necessary improvements, I am confident the paper has the potential to make a valuable contribution to the field.

I recognize, from first hand experience, the amount of effort that went into this work. This is why I want to restate my strong support for the manuscript as I think it brings an important contribution to the field. I strongly encourage the authors to address the points I have highlighted, as doing so will enhance the clarity, rigor, and overall impact of the work.

Greg Guillet, University of Oslo

**General Comments**

- From this version of the manuscript it is not clear at all to me how uncertainties are calculated, nor what do they actually represent. More details on this throughout the manuscript are severely needed.

- Section 4 needs a vocabulary overhaul.
  The authors use very unspecific language which makes it harder to grasp what is the point they are trying to make. I have addressed some of these in my specific comments.

- As it stands, the writing in Section 5 makes it very hard to understand. I had to re-read some sentences multiple time to make sure understood the statements correctly. I think significant efforts are needed to

- There are some problematic statements which show some confusion between GPs and splines. I think these can and should be addressed with minimal changes by removing the statements I have highlighted in my specific comments.
  In addition, the authors need to reframe comparisons between methods in the manuscript as such : comparisons between the method proposed by the authors and the GP from Hugonnet et al. (2021).

- A lot of the sentences start with unspecific pronouns such as "this" or "it" - which required me to backtrack a few times. This severely hampers the readability of the manuscript.

- Acronyms are not always defined on their first use - please check this.
- Finally, there are quite a lot of frenchisms and typically french sentence constructions - it's not a major problem, but the manuscript would gain in readability if the authors addressed this.

**Specific comments**

**Abstract**

> We compare the produced dataset to previous studies [...]

If I am not mistaken, you only compared it to the Hugonnet et al. (2021) results.
I would hence keep it singular here.

**Introduction**

I generally agree with all the statements made in the introduction and they are accurate.
However, as it stands, I find the structure of this introductory section quite confusing.
The authors often switch between very broad statements to nichely precise descriptions of methods/problems without clear guidance of the reader into why this matter, which ultimately leads to significant repetition.
This is especially true between lines 30 and 50, which is followed by a lengthy statement on glacier surface velocity inversion methods which seems a bit out of place - I would either make the point clearer as to why this is important here or remove it.

Within these lines, I suggest a little restructuring :

- What are surges and why are they important (essentially your lines 16 to 23)
- Why are elevation changes particularly important when studying surges (which is basically your lines 35 to 34) but I would focus on 1) visual interpretation (i.e. mostly inventory efforts) and 2) quantitative interpretation - i.e. computation of shear stress, and even more, ice thickness/bed geometry inversions. I would remove the mentions to velocity products
- Problems with the data and then the problems with the methods used up to know and how you're solving them.

> [...] are not yet fully understood and this subject continues to be the object of developments and theories (Benn
> et al., 2023; Terleth et al., 2021; Thøgersen et al., 2024; Crompton et al., 2018)

I suggest to keep citations in chronological order, and to use e.g. in front of the citation array here, as these are but a subset of very relevant publications addressing the topic.

> L37: benefits

Benefit.

> L39: The use of suitable digital elevation models (DEMs)

You already defined used DEM on line 38, define it there.

> L40: suitable

I am not sure I grasp what this means in this context, is it DEMs that have high enough accuracy ? Please clarify.

> L49: They need techniques [...]

In general, I suggest avoiding sentences starting with "they" and would try to rephrase.

**Paragraph starting at line 52**

May I suggest adding my own work - Guillet and Bolch, 2023 - where we develop a Bayesian outlier filtering and uncertainty quantification framework to compute thickness changes from DEMs, specifically for surge-type contexts.
While we do not specifically tackle dense time series, the methods would be the same for denser data arrays and akin to data assimilation.
This is barely a suggestion, and I leave the decision to add this reference to the authors.

- Guillet, Gregoire, and Tobias Bolch. "Bayesian estimation of glacier surface elevation changes from DEMs." Frontiers in Earth Science 11 (2023): 1076732.

> L65: "To accurately estimate the parameters"

What does "parameters" refer to here ? I assume this is thickness change/dynamical thickening but be precise.

**Paragraph starting at line 68**

I would suggest to bolster this paragraph and be a bit less succinct - you have done a lot of work here and this is a good place to showcase a short summary.

> We algorithms from the literature to filter outliers

This reads awkward. I suggest changing for "We use established algorithms [...]"

**Section 2**

**Paragraph starting at line 80**

There are several sentences starting with "they" in a rapid succession, which hampers the readability of the whole paragraph. Please rephrase.

> L92 : The temporal sampling is heterogeneous in time and space.

Rephrase.

> L92-95

It took me 2 reads of these sentences to understand what was clearly meant. First, "below 50 days apart" should be "less than 50 days apart", but a more general question is whether both sentences are needed. I feel they give redundant information, that is illustrated in Figure 2.
I would rephrase to avoid this repetition and clarify the statement.

> L 96-97: (European Space Agency and Airbus, 2022)

Reference should be after the mention to the DEM.

**Section 3**

**Section 3.1**

> 1. Spatial filter: we filter out pixels with a difference of more than 400 m between the ASTER
>    DEM and GLO-90 reference
>    DEM.
> 2. Merging of strips: we merge the DEM strips on the same day by keeping, at the pixel level,
>    the elevation with the highest
>    correlation score at overlaps.

1. Is this 400m threshold arbitrary ? It's not a problem if it is - I am wondering if there is a reference to back this up or if its from the author's personal experience with the data.
   Also, the notion of "spatial" filter is a bit confusing as you are filtering pixels that show and absolute difference in elevation ($z$ coordinate) between the ASTER and GLO-90 DEMs, correct ? Or are you operating in the $x, y$ plane ?
2. This will probably reveal my total lack of knowledge on DEM generation methods but, is the correlation score a sufficient enough metric to discard data ?
   Or is there value in computing a median DEM from all the available DEMs for this day and using this instead ?

> L111-117

This could be condensed a bit.

**Section 3.2**

> L125: For our dataset, the output of the regression is to sensitive [...]

too sensitive

> L 125-126 : For our dataset, the output of the regression is to sensitive to noise overall and too smooth over surges to be used directly as an interpolation of the elevation, so we use it for

> filtering only.

This is a pretty important point which I feel is a bit brushed over by the authors here, as it is pretty well documented that LOWESS struggles in non-stationary contexts.

**Section 3.3**

> L147: "thereby improving the state of the art in this domain"

I suggest to remove this. This is very vague and does not give any useful information.

> L165-170.

I have to say I strongly disagree with this statement and I think there is a bit of confusion that needs to be addressed.
While GPs do require a prior on the functional form to fit a given dataset, the shape of the kernel function can actually be interpreted as physical variations in the latent variable, i.e. the shape of the Linear + Periodic kernel used in Hugonnet et al (2021) reflects the sub-linear and periodic signal one could expect from surface elevation changes.
Framed as a causal inference problem, a GP should not be used with prior belief on the structure of the data, but on the physical process the data represents.

ALPS also puts a "prior" on the functional form to fit, as it relies on a combination of b-splines to adequately approximate local regions of the data.
Even if you were interpolating with linear functions, you would be making an assumption that there is no oscillatory behavior between data points - which, by definition is a prior assumption.
The main difference here is thus that the "prior-like" assumptions are expressed through, for example, the smoothness of the derivative (splines) instead of a covariance function (GP).
In a more general way, any model requires prior assumptions.

As a side note, I want to point out that standard smoothing splines represent a special case of GPs, as shown by Kimeldorf and Wahba (1970).
In a highly-abstract way, you are, implicitly, fitting different GP models to different regions of the dataset - one should not stretch this analogy too far, as it breaks down when considering uncertainty estimates, as GPs are probabilistic.

All in all, I would argue against the generalization of this statement to every GP model, and only focus on the one defined by Hugonnet et al. (2021), otherwise this is not an apple-to-apple comparison.
You are comparing the results of two different methodologies and how they fare at interpolating time series of surface elevation data in the presence of transient physical events, no more, no less.
In addition, and I think this point is an important plus for the author's approach : I would imagine ALPS is more scalable than any GP model over larger sample and dimensions datasets, as GP are knowingly computationally expensive.
This is a strong pro for your method and I think you should address a bit more, as surface elevation time series are likely to get denser in the future.

- Kimeldorf, George S., and Grace Wahba. "A correspondence between Bayesian estimation on stochastic processes and smoothing by splines." The Annals of Mathematical Statistics 41.2 (1970): 495-502.

**Section 4**

**Section 4.1**

> L188-189: Modifications of this kernel to allow
> for stronger changes in elevation have not proven to be efficient enough

I would be a bit more careful and specific here, as I know of successful attempts at this with different GP kernels.
I would write something of the like : "Reparametrization of the kernel used by Hugonnet et al (2021) [...]".
It seems nitpicky, but I think it makes your point stronger.
Also be careful when using "efficient" - I get that you mean that the interpolation result is not satisfactory or rather than the median is too far from what you would expect, but efficiency is a very vague term.

> L 190-192: It does conserve nearly all known surge events in our study area and period, with
> one exception being surge events with strong melt before and after the surge

When you say study area and period, you mean the 4 surges you look at, correct ?
In which case, I would rephrase this as I think in its current form, it is not really reading well : "nearly" is a rather imprecise formulation that could be interpreted as trying to hide the cases where it does not work so well.
Just say 3 out of 4. Then, "events" is singular.

> L193: smooths

Smoothes

> L197-203

This whole section is confusing.
I would remove "unfavorable terrain" and be a bit more specific - directly mention textureless and steep regions.
"unrealistic erratic" is also confusing, I get what you try to mean here, I would use something like "Unstable" or "Oversensitive to Noise".
"The filtered-out areas (data gaps) are more prevalent with our method, mostly over unfavourable terrain": more prevalent with your methods than what ? The GP from Hugonnet et al (2021) I assume ? Be more specific here.
What I get from this section is that your method works better than Hugonnet et al (2021) on-glacier, in parts with that are relatively smooth, but tend to over-filter in areas of low contrast/rough terrain - am I correct ? In any case please make the section easier to understand.

> L205

Again, this is confusing. Please rephrase.

**Section 4.2**

> L209-212;

An interesting point is that the GP used in Hugonnet et al (2021) shows an increasing trend in surface elevation, completely omitting the actual data.
GPs tend to "fall-back" to the median when there is no data but here, both the median and the uncertainty increase.
Can you plot the uncertainty of each measurement ?

> L211: undulations

Replace with "periodic component"

> L215: "[...] creating wavelet artifacts"

The term wavelet design something different in signal processing and I would refrain from using it here. I would use "spurious high frequency oscillations" or something similar.

> L216: "removes completely the surge
> signal"

filters the surge signal out

> L218: have weakest changes

Weaker. But please consider changing to a more specific term.

> L220-221: Some glaciers are more affected by data gaps than
> others, in agreement with areas with a low number of observations (Fig. 1, e.g. Shisper
> glacier)

Shisper is not highlighted on Figure 1 and is not in the studied glaciers.

**Section 4.3**

> L227: with a decreasing speed (2009-2012, a1),

This figure does not show the velocity.
Also, the reference to the figure is broken.

> L235: wavelets

Same as before.

> L241: to the end of the surge, it then extends 5-10 more

to its termination, the surge propagates

> L246:

Add uncertainty estimates - at least some part of the discrepancy is in there.
I imagine there is an underestimation in the surface area of the reservoir zone ? Did you account that the surge also simultaneously drains the northern (Yutmaru?) tributary (centerline RGI2000-v7.0-L-14-27499) ?

> L253 :The "build-up front" or kinematic wave

To stay consistent with current terminology I would use surge front, not build-up front or kinematic wave.

> L254-255: representing a regular advance of about 460 m per year, which is approx-
> imately 6 times faster than the surface velocity, according to the NASA MEaSUREs ITS_LIVE
> project repository

I am not sure I get what you mean here - the surface velocity data at Khurdopin clearly shows seasonal behavior with velocities reaching around 400-450 $\mathrm{m} * \mathrm{yr}^{-1}$, starting in 2013 with a quasi-linear increase in velocity up to 2017.
Although I might be wrong, i would expect you to be able to see that the surge front advance rate is slower between 2000-2012 than 2012-2017 when the glacier slowly starts to shift to a velocity weakening regime.

> L275-76: The buildup and emptying of the first surge seems weaker than the second one, and
> extends less up-glacier of the junction, compared to the second surge

Again, refrain from using weaker as it gives the false idea that the surge did not dissipate as much energy - something we have no idea on.
The peak velocities of both of Yazghil surges are actually pretty similar and both are visible up to the glacier front in the surface velocity record.

> L277-278: This may be related to the effect of the tributary surge, that stopped at the junction
> but could have yet increased mass input by a blocking effect.

I really don't get what you mean by that, please explain.

**Section 5**

**Section 5.1**

This whole section is very confusing.

I do not understand what the first sentence is supposed to mean, how can an uncertainty estimate over a quantity reflect the filtering capabilities of a filter ?

What does it mean that the surge of Khurdopin shows that "that a discrepancy of a hundred meters is credible on exceptional events." ?

In addition, to further test the outlier filtering side of your methodology, you could generate false erroneous measurements and further quantify how well your method performs at filtering simulated outliers.

> L281: keep true elevations

Be careful with the use of "true".
All measurements are imperfect representations of the "true" elevation, which is by definition, unattainable.

> L309

Add a full stop before "To test"

**Section 5.2**

> L332 : appears

Appear

> L337: pre-surge thickening front or kinematic wave

Same as before. The propagation of a thickening front is one of the definitions of a surge. In your case it is still the early stage and has not reached the dramatic proportions it will eventually attain.
Stop using kinematic wave.

> L341: There is

There are

> L343: a bit later than our spring 2016 estimate

Unspecific. If it's spring versus October, then it's around 4/6 months later, just say it.

> L360-370:

Be a bit more specific. Please add uncertainty estimates. Mention that differences you show are between median values

> L376: one of the shortest surge cycles in HMA.

Is this from Bhambri et al. (2017) ? Make sure to add proper reference

> L378: Our data suggest it started 1-2 years later, implying a longer quiescence phase of 11-13 years

Do not make this a general statement on the dynamics of Yazghil glacier - $\approx 8$ years of quiescence is not different from $\approx 11$ when the number of considered events is 2.

> L385: The order of magnitude of the imbalances corresponds to the order of magnitude of the measurement uncertainty

Can't agree more ! Please add them.

> L383: bulge front

Just use bulge. Also, as mentioned before, it's not pre-surge.

**Section 5.4**

Typos

> L421: (exponential sine-squared (ESS) kernel)

I doubt being so specific is really needed here.
Just remove it. Also, components of a kernel are traditionally called "terms".

**Conclusions**

> L449-451

Succession of sentences starting with "it"

**Figures**

In general, all captions need to be reworked to describe all individual panels of the figures. I provide more specific comments hereafter.

**Figure 1**

Readers familiar with surges and HMA will know where the glaciers you mention are, I am not sure this is the case for the broader audience - maybe you could zoom in on a bounding box around the selected glaciers.
I am not sure the whole Kararokam region needs to be displayed since you focus on specific glaciers.

**Figure 3**

Caption : I cannot find any mention to "TS" in the plot - remove this from the caption since you write time series.

It would be beneficial to know which glacier centroid/vertex this surface elevation time series is sampled from.

This is a bit of a nitpick here but I would refrain from using two similar colors for the lines in "Interpolation it. 1" and "Interpolation it. 2" - being colorblind, I can't see the difference between them.

In addition, it would be beneficial if you showed the uncertainty associated to each measurement on the plots to the right.

Finally, I see no mention to any Student-T distribution in the paper (because the methods you rely on make no explicit assumption on the distribution of the data). Rename the "t-interval" into "Confidence Interval".

**Figure 6**

It's really hard for me to see the individual points between raw elevation and filtered measurements. I think the symbols and the figure in general are just too small.
Again, just remove t-confidence interval and use confidence interval. i think it's too specific for most readers - if they want to know more, they will read Shekhar et al. (2021).

**Figure 7**

Increase the size of the figure and individual panels.

**Figure 8**

Hovmöller diagrams are a widely used plot in the community.
Just use "Interpolated surface elevation time series along the centerline of..." or something of the sort.

**Figure 9**

Again, just specify which glaciers these time series are sampled from - and where on the glacier.

**Figure 12**

Please add the red A, B and C regions in the captions. It's a shame to have to go into the text to grasp what the figure shows.

**Tables**

Please add uncertainty estimates in all tables - as a hunch feeling I would typically assume that this is what drive the reported imbalance (except for Hispar).

**Table 2**

This table is pretty confusing.
I would suggest replacing Table 2 with a figure showing the distributions for each glacier.
This would avoid having 2 columns as the 90th percentile and show the full distribution.

---

## Author Comment (AC1)

**Response to the comments from William Kochtitzky, reviewer #1.**

The authors present a new method of computing elevation changes during surge events when numerous elevation measurements are available for glaciers in a relatively short time period (annual to decadal). The method is novel and advances our knowledge of surging. They provide new insight into several different surge events that have been previously documented. I recommend the paper be published, but I have a few minor comments that could strengthen the paper below. My main criticism is that the authors seem to lack a quantified uncertainty of their results. For example, it would be greatly beneficial to add uncertainty to table 1. I don't think another round of review is necessary, I am happy to see it published after the authors make these minor changes.

-Will Kochtitzky

We thank Will Kochtitzky for their careful reading of our manuscript and their comments. We have made all the minor changes suggested, please find below the remaining points addressed. Our answers are in blue.

- Figure 2 – very cool figure, but I am having a hard time understanding it. I think part of the problem is that I don't get the colorbar showing the density of DEMs. I don't see this mentioned in the caption. Is it showing how much of each DEM is good data? Maybe don't make the line and the colorbar blue to add clarity – the caption is confusing which blue you are referring to.
    Thanks for the positive feedback. The information given by the 2D histogram / density of DEM was not very relevant and difficult to understand, we choose to removed it.

- Page 5 – can you give us a sense of the data that you filtered out? What is the percent of the data that was filtered out of your study?
    We added a sentence to the last paragraph of the subsection 4.1 Results / Performance of the outlier filtering : "The pre-filtering step removed 46% of the original regional dataset (number of on-glacier pixel), and the filtering step removed a further 42% of the pre-filtered dataset (69% removed in total compared to the original dataset).".

- Figure 7 caption – you say in the text what the red circles are, but this should also be added to the caption for clarity:
    We completed the caption with the following sentences : "The red circles and the dotted lines show or delimitate areas discussed in the text.". We may further add labels.

- Line 245-250 – this is very interesting – do you think you are not capturing the mass fully? Where could it be coming from? Are parts of the reservoir zone not included in your calculations?
    There are indeed parts of the glacier that are not included in the calculation, please find attached a map showing in thick black lines the manually drawn reservoir and receiving areas (Fig. R1.1). However, a similar imbalance is computed when using the full outline (red on the figure) : -3081 .$10^6$ m³ and 3956 .$10^6$ m³ in the reservoir and receiving areas, respectively. We completed a sentence to add this information. The imbalance varies with glaciers and dates chosen (Table 1).
    We did not analyse enough surge events in this paper to draw generalities, and we do not have explanations for this. We extended significantly the discussion about crevasses in discussion, subsection 5.2: "The difference between volume gain and loss we estimate is equivalent to a layer of 4.55 m thickness over the surge area. This imbalance is unexpected as the surge occurs over a short time period and mass should be roughly conserved. The imbalance is quite similar when using two filtered ASTER DEMs over a similar period over this surge, instead of the interpolated series, or when calculated over the full glacier system instead of over the delineated reservoir/receiving areas. The impact of crevasse opening during the surge on the apparent surface elevation has not been assessed, especially regarding our imbalance, but it may represent a non-negligible volume. The opening of crevasses can be equivalent to up to 0.2 m thickness over regional scale of the Greenland Ice Sheet (Chudley et al., 2025). As inland parts of

these regions are largely crevasse-free, we can expect such volume to be significantly larger over the highly crevassed post-surge surface of Hispar glacier, at least one meter magnitude. By mid-2018 our imbalance is equilibrated, as well as is the imbalance of Bhambri et al. (2022) with a end term in 2020 when a number of crevasses have already closed. The Khurdopin and Kyagar glaciers were already crevassed a lot before the surge, and this effect may be less important"

[Figure]

Fig. R1.1.Map of the elevation change over the Hispar 2014-2016 surge and boundaries used for our calculation.

- Table 1 – what are your uncertainties on these measurements? This is critical since any imbalance outside the uncertainty would be a more important signal. Can you get these like you did for figure 6?

   We understand the concern of the reviewer and acknowledge that our manuscript was not hundred percent clear regarding uncertainties. Below we detail a number of uncertainties that we clarified in the revised manuscript.

   - On the figure 6, 3 and 9, we show confidence intervals that are those of the interpolation algorithms (GP Regression and ALPS) and not of the overall workflow. We do not think that the theoretical uncertainty and confidence interval predicted by the ALPS-REML are credible to assess the results of the overall workflow. For instance, Figure A2.a (in appendix) shows well that the filter can remove some critical portions of the surge signal (creating data gaps that could also exist in the original dataset) without the interpolation uncertainty to represent it. A visual verification of the process permits to detect large biases, but does not give any metric value. This is why we develop a lot the sensitivity analysis, the comparison with external reference DEMs or reference study on a number of events. It permits to give some sense of the uncertainty, of the ability and weaknesses of the method.
   - Still, for the revised version of our manuscript, we will give some uncertainties on the surge volumes (e.g. in Table 1). These uncertainties are based on standard methodologies based on the elevation change from stable terrain used as a proxy to estimate the uncertainty on moving terrain (Hugonnet et al., 2022). The first estimations are very conservative and gives uncertainties that, on a median basis, are about 55% of the estimated volume of ice transferred (with volumes of Table 1 of the manuscript; from 19% to 280%). We will try more developed uncertainty calculations and complete the revised version accordingly.

- Figure 10 – shouldn't the colorbar be elevation difference? Change implies the difference is real, but if I understand this correctly, the difference should be 0.

    This is right and is now corrected.

- Line 330 – are they within uncertainties?

    As we do not provide uncertainty on our volume transfer estimates yet (see answer above), we cannot fully answer this question. However, our estimate in the receiving area is close to the uncertainty range of Bhambri et al. (2022) and it would likely be within uncertainties. According to our first estimate of our volume uncertainty, it is indeed largely within uncertainties, but they are very conservatives. For notice, we cannot retrieve the uncertainty of Bhambri et al. (2022) for the reservoir area estimate, as we cannot retrieve the uncertainty they allocate to the Khani Basa tributary.

- Line 338 – the main sentence on this line is grammatically incorrect, I am not sure what you are trying to say

    This now reads as "The existence of kinematic wave that propagates the surge front have regularly been observed on other surges" (previously, "kinematic wave propagating the surge front").

- Figure 12 – need to add what areas A, B, and C mean to the caption

    We completed the caption with "The areas and trends designated in red are discussed in subsection 5.3. They highlight areas of large surge smoothing or removal (zone A) or overall smoothing of elevation changes (trend B) by the original method (Hugonnet et al., 2021), and artefacts created by the presented workflow (zone C).".

---

## Author Comment (AC3)

**Response to the comments from Gregoire Guillet, reviewer #3.**

The manuscript proposed by Beraud et al. presents a new methodology to filter outliers and estimate glacier surface elevation from dense digital elevation model time series.
Their study relies on the same data as the Hugonnet et al (2021) study.
Instead of Gaussian Process Regression, Beraud et al implement an additional LOWESS filtering step, before feeding the fitlered time series to a localized b-splines scheme in order to interpolate surface elevation measurements.
While the work and the methods shows promise, I strongly recommend that the authors take the necessary steps to strengthen the manuscript.
The paper appears to be in an early stage of development and would benefit from significant revisions before it is considered for publication: sections 4 and 5 are particularly challenging to understand. With the necessary improvements, I am confident the paper has the potential to make a valuable contribution to the field.
I recognize, from first hand experience, the amount of effort that went into this work. This is why I want to restate my strong support for the manuscript as I think it brings an important contribution to the field.
I strongly encourage the authors to address the points I have highlighted, as doing so will enhance the clarity, rigor, and overall impact of the work.

Greg Guillet, University of Oslo

We thank Gregoire Guillet for their careful reading, their comments and help in improving our manuscript.
We considered all the comments. The minor ones have been accepted and do not appear in this response document. Please find below our answer and larger changes in response to other comments. Our answers are in blue.

**Answers to general comments**

- From this version of the manuscript it is not clear at all to me how uncertainties are calculated, nor what do they actually represent. More details on this throughout the manuscript are severely needed.

    We understand the concern of the reviewer and acknowledge that our manuscript was not hundred percent clear regarding uncertainties. Below we detail a number of uncertainties that we clarified in the revised manuscript.

    - On the figure 6, 3 and 9, we show confidence intervals that are those of the interpolation algorithms (GP Regression and ALPS) and not of the overall workflow. We do not think that the theoretical uncertainty and confidence interval predicted by the ALPS-REML are credible to assess the results of the overall workflow. For instance, Figure A2.a (in appendix) shows well that the filter can remove some critical portions of the surge signal (creating data gaps that could also exist in the original dataset) without the interpolation uncertainty to represent it. A visual verification of the process permits to detect large biases, but does not give any metric value. This is why we develop a lot the sensitivity analysis, the comparison with external reference DEMs or reference study on a number of events. It permits to give some sense of the uncertainty, of the ability and weaknesses of the method.

    - Still, for the revised version of our manuscript, we will give some uncertainties on the surge volumes (e.g. in Table 1). These uncertainties are based on standard methodologies based on the elevation change from stable terrain used as a proxy to estimate the uncertainty on moving terrain (Hugonnet et al., 2022). The first estimations are very conservative and gives uncertainties that, on a median basis, are about 55% of the estimated volume of ice transferred (with volumes of Table 1 of the manuscript; from 19% to 280%). We will try more developed uncertainty calculations and complete the revised version accordingly.

    L360-370:

Be a bit more specific. Please add uncertainty estimates. Mention that differences you show are between median values
L385: "The order of magnitude of the imbalances corresponds to the order of magnitude of the measurement uncertainty"
Can't agree more ! Please add them.
Tables: Please add uncertainty estimates in all tables - as a hunch feeling I would typically assume that this is what drive the reported imbalance (except for Hispar).

> Our answer to the general comment covers these three minor comments above. We will give some uncertainty estimates in the revised version, reminding it does not include all kind of bias (e.g. for the wrong processing of the Khurdopin surge).

- Section 4 needs a vocabulary overhaul. The authors use very unspecific language which makes it harder to grasp what is the point they are trying to make. I have addressed some of these in my specific comments.
  As it stands, the writing in Section 5 makes it very hard to understand. I had to re-read some sentences multiple time to make sure understood the statements correctly. I think significant efforts are needed to
  A lot of the sentences start with unspecific pronouns such as "this" or "it" - which required me to backtrack a few times. This severely hampers the readability of the manuscript.
  Acronyms are not always defined on their first use - please check this.
  Finally, there are quite a lot of frenchisms and typically french sentence constructions - it's not a major problem, but the manuscript would gain in readability if the authors addressed this.

  > The overall article and these sections particularly will be proofread and improved for the revised version. The specific comments are addressed in this answer.

- There are some problematic statements which show some confusion between GPs and splines. I think these can and should be addressed with minimal changes by removing the statements I have highlighted in my specific comments. In addition, the authors need to reframe comparisons between methods in the manuscript as such : comparisons between the method proposed by the authors and the GP from Hugonnet et al. (2021).

  > Regarding the confusion between GPs and splines, we answer directly to the specific comment below in this document.
  > About clarifying the different comparisons: we completed the end of the introduction with "We evaluate the performance of the workflow compared to the results of Hugonnet et al. (2021). We also compare the surge characteristics such as volumes transferred to other products and studies *[N.B.: studies of the literature, that worked on the case studies of a few surge events]*.". We also already insist on the fact that we compare a lot our workflow to the GP regressions as implemented by Hugonnet et al. (2021). The first paragraph of the discussion subsection 5.4 Methodological Insights and Modifications states that our comparison does not apply to all possible settings of GPR.

**Answers to specific comments**

- "We compare the produced dataset to previous studies […]".
  If I am not mistaken, you only compared it to the Hugonnet et al. (2021) results. I would hence keep it singular here.

  > With this sentence, we also refer to the work done to compare the four surge events we analyse with the dataset produced. In the subsection 5.2 Discussion / Comparison of surge characteristics with the literature, we compare dates, volume transferred etc. against other studies (e.g., Bhambri et al. (2022), Steiner et al. (2018), Gao et al. (2024)...).

- I generally agree with all the statements made in the introduction and they are accurate. However, as it stands, I find the structure of this introductory section quite confusing. The authors often switch between very broad statements to nichely precise descriptions of methods/problems without clear guidance of the reader into why this matter, which ultimately leads to significant repetition.

[...]

*We reorganised the introduction, with a few minor changes of content.*

- Paragraph starting at line 52 - May I suggest adding my own work - Guillet and Bolch, 2023 - where we develop a Bayesian outlier filtering and uncertainty quantification framework to compute thickness changes from DEMs, specifically for surge-type contexts.

  *Thanks for this suggestion. We included this work by these few lines : "A recent study has exploited a Bayesian framework by inference applied to elevation change to filter outliers, which requires prior knowledge from diverse sources (Guillet and Bolch, 2023). It has been tested on surge-type glaciers, and it applies equally to dense time series.".*

- Paragraph starting at line 68, I would suggest to bolster this paragraph and be a bit less succinct - you have done a lot of work here and this is a good place to showcase a short summary.

  *We completed 3 sentences to add a few details. This now reads as "In this study, we present a workflow designed to filter and interpolate elevation time series of high temporal resolution during surge events. We use established algorithms to filter outliers and interpolate elevations at monthly scale while preserving surge elevation signals. We apply it to an unfiltered ASTER DEM dataset from Hugonnet et al. (2021). We produce a regional dataset in the Karakoram region covering more than 100 surge-type glaciers. We evaluate the performance of the workflow compared to the results of Hugonnet et al. (2021). We also compare the surge characteristics such as volumes transferred to other products and studies."*

- L92 : The temporal sampling is heterogeneous in time and space. Rephrase.

  *This now reads as "The temporal sampling is not regular in time, and parts of the mountain range have about twice less DEMs than others".*

- It took me 2 reads of these sentences to understand what was clearly meant. First, "below 50 days apart" should be "less than 50 days apart", but a more general question is whether both sentences are needed. I feel they give redundant information, that is illustrated in Figure 2. I would rephrase to avoid this repetition and clarify the statement.

  *We removed the first sentence.*

- Is this 400m threshold arbitrary ? It's not a problem if it is - I am wondering if there is a reference to back this up or if its from the author's personal experience with the data.
  Also, the notion of "spatial" filter is a bit confusing as you are filtering pixels that show and absolute difference in elevation (z coordinate) between the ASTER and GLO-90 DEMs, correct ? Or are you operating in the x, y plane ?

  *Yes the 400 m threshold is arbitrary. Some surges with up to 200 m elevation change have been observed and we do not know any observation with more than this value, our threshold is thus quite conservative.*
  *The name is indeed not appropriate, thanks, we now call it "absolute filter".*

- This will probably reveal my total lack of knowledge on DEM generation methods but, is the correlation score a sufficient enough metric to discard data ?
  Or is there value in computing a median DEM from all the available DEMs for this day and using this instead ?

  *The reviewer is correct that the correlation score is not always the best metric to exclude some data, we write in the subsection 4.1 Results / Performance of the outlier filtering: "This is often due to scattered elevations and to the correlation score that is not very representative of the actual pixel quality: outliers may have lower uncertainties than more accurate observations (e.g., Fig. A2.e).". Here below is the figure (Fig. R3.1), showing some points of various elevation error (representative of the correlation scores for a pixel time series) mixed altogether :*

[Figure]

Fig. R3.1. LOWESS workflow over a sample time series.

About merging stripes based on the maximum correlation score versus a median of values, yes both are possible. A few specific cases where in favour of using correlation score. However, overlaps represents a small portion of the data, so the impact should be very limited.

- L111-117 This could be condensed a bit.
    We removed 2 sentences out of these 6.

- L 125-126 : "For our dataset, the output of the regression is to sensitive to noise overall and too smooth over surges to be used directly as an interpolation of the elevation, so we use it for filtering only."
  This is a pretty important point which I feel is a bit brushed over by the authors here, as it is pretty well documented that LOWESS struggles in non-stationary contexts.
    We added the sentence "indeed, regressions struggles in non-stationary contexts (Baumohl and Lyocsa, 2009)."

- L165-170. I have to say I strongly disagree with this statement and I think there is a bit of confusion that needs to be addressed.
  While …
    We thank the reviewer for their explanations, and acknowledge that our text was not correct about these methods. My earlier view on B-splines was simplistic with a "user" point of view and I was lacking the theoretical background. Although we keep a very succinct level compared to reviewer's explanations, the modified paragraph now reads as:
    "We compare Gaussian Process Regression (GP regression) from Hugonnet et al. (2021) and ALPS-REML in our study. GP regression, equivalent to kriging, is a non-parametric method that relies on estimating the data covariance to provide an optimized interpolator. Under certain assumptions, including notably second-order stationarity, GP regression has been shown to be the "best linear unbiased predictor". It is the method used by Hugonnet et al. (2021) on this same dataset, to compute long-term mass balance estimations worldwide.  We use a GP covariance with terms estimated in Hugonnet et al. (2021) through a global variogram analysis. This analysis identified several kernel components (periodic, local, linear...), that  are not specifically tuned for surges. B-splines and thus ALPS, on the opposite, approximate the data with polynomials under the assumption of a degree of smoothness of the data, with no need for us to inform the behaviour of the data. Although both GP regression and ALPS need domain knowledge to decide the covariance kernel and spline degree/penalty respectively, from a user's perspective using GPs is harder owing to the well studied difficulty of finding the right kernel (Pu, 2024). For ALPS on the other hand, we just need to choose the right degree and penalty order from a small set of choices, and that results in a more stable fit in our case.".

- L188-189: Modifications of this kernel to allow for stronger changes in elevation have not proven to be efficient enough I would be a bit more careful and specific here, as I know of successful attempts at this with different GP kernels.

This now reads as "Reparametrization of the kernel used by Hugonnet et al. (2021) gave slightly worse results than those obtained with the ALPS-REML method.". Indeed, the reparametrization gave fairly satisfactory results, and it is likely that a better-defined kernel could even outperform the results of the ALPS-REML method.

- L 190-192: It does conserve nearly all known surge events in our study area and period, with one exception being surge events with strong melt before and after the surge When you say study area and period, you mean the 4 surges you look at, correct ? […]

    It also applies to a broad number of surge events in the Karakoram, although we did not check manually each surge-type glacier. We rephrased our statement this way : "It conserves well the surge signal of 3 out of the 4 events we analyse in subsection 4.3, and this observation seems to extend to a number of surge events in Karakoram. One exception is the situation of surge events with strong melt before and after the surge.". Another case of failure for example is at the front of the Aktash glacier (RGI2000-v7.0-G-14-18524).

- L197-203 This whole section is confusing. […]
  What I get from this section is that your method works better than Hugonnet et al (2021) on-glacier, in parts with that are relatively smooth, but tend to over-filter in areas of low contrast/rough terrain - am I correct ? In any case please make the section easier to understand.

    We applied the minor changes suggested and we simplified some parts of the subsection. We added the following sentence at the beginning of the last paragraph : "To summarize, our filter permits to preserve better the surge signals that were filtered out in the workflow of Hugonnet et al. (2021). However, the new filter is more noise-sensitive over textureless accumulation areas and rough terrain, leading to data gaps or artifacts with large elevation changes.".

- L205 Again, this is confusing. Please rephrase.

    This sentence and the one at the end of the section 2 "Data" have been thought through quite much during the writing process already, without any simple solution. We could reason by probability, which we did not keep: "Any random date in the time series period have 40% (75%, respectively) probability to fall between unfiltered observations less than a year (two years, respectively)". We will consider the thoughts of the reviewers on this on a possible second round of review.
    As a reminder, the current version is "After filtering, nearly 40% (75%, respectively) of any date in the time series periods are between unfiltered observations less than a year (two years, respectively)".

- L209-212; An interesting point is that the GP used in Hugonnet et al (2021) shows an increasing trend in surface elevation, completely omitting the actual data. GPs tend to "fall-back" to the median when there is no data but here, both the median and the uncertainty increase. Can you plot the uncertainty of each measurement ?

    We show on the figure below (Fig. R3.3) the time series coloured by the elevation error (in m), which is estimated with the workflow from Hugonnet et al. (2021). We added the sentence "It is noteworthy to mention that by its design, the original kernel is optimized to preserve a linear trend to extrapolate out of the observation period of each pixel.".

[Figure]

Fig. R3.3. Time series with the interpolation of Hugonnet et al. (2021) and the per-pixel elevation error estimates.

- L215: "[...] creating wavelet artifacts", The term wavelet design something different in signal processing and I would refrain from using it here. I would use "spurious high frequency oscillations" or something similar.
    We thank the reviewer for the appropriate suggestion, we replaced our old term with it.

- L218: have weakest changes, Weaker. But please consider changing to a more specific term.
    We changed to "show smaller elevation changes"

- L220-221: "Some glaciers are more affected by data gaps than others, in agreement with areas with a low number of observations (Fig. 1, e.g. Shisper glacier)", Shisper is not highlighted on Figure 1 and is not in the studied glaciers.
    Showing Shisper glacier was not intended, it was just an additional piece of information because our dataset did not allow us to add its surge to our panel due to large data gaps, even though it is a well-studied case. For clarity, we removed the mention of Shisper glacier.

- L227: "with a decreasing speed (2009-2012, a1)", This figure does not show the velocity. Also, the reference to the figure is broken.

We here talk of the surge front propagation, which is visible on the Hovmoller diagram, we do not talk about ice velocity. To avoid further confusion, we replaced "speed" by "rate of propagation". The reference to the figure has been corrected.

- L246: Add uncertainty estimates - at least some part of the discrepancy is in there. I imagine there is an underestimation in the surface area of the reservoir zone ? Did you account that the surge also simultaneously drains the northern (Yutmaru?) tributary (centerline RGI2000-v7.0-L-14- 27499) ?

  As discussed previously, we do not provide uncertainty on interpolated elevations (and thus on volume transfer estimates) yet, but we will add some in the revised version. A first conservative estimate of the the uncertainties for the transferred volume are : -2411 ±1290 .$10^6$ m$^3$ (reservoir area) and 3110 ±605 .$10^6$ m$^3$ (receiving area).

  Indeed some parts of the discrepancy should lies in the area delineations, although the choice of dates could have an even greater impact (we state that "By mid-2018 [end term of the DEM difference] our imbalance is equilibrated").

  Yes we include the northern/Yutmaru tributary. Here below (Fig. R3.4) we show the elevation map with the distinct reservoir and receiving areas (in black) we used for calculating the surge volume. We have a similar imbalance when we compute it at the scale of the glacier system (the whole red outline, including Kunyang tributary).

[Figure]

Fig. R3.4. Map of the elevation change over the Hispar 2014-2016 surge and boundaries used for our calculation of transferred volume.

- L253 : "The "build-up front" or kinematic wave"; To stay consistent with current terminology I would use surge front, not build-up front or kinematic wave.
  L337: "pre-surge thickening front or kinematic wave"; Same as before. The propagation of a thickening front is one of the definitions of a surge. In your case it is still the early stage and has not reached the dramatic proportions it will eventually attain. Stop using kinematic wave.

  Thanks for rising this point. It is indeed different from a kinematic wave, although for example Turrin et al. (2013) use both terms ambiguously in a similar situation. We modified extensively several paragraphs of this section for the sake of clarification and revised the terminology. Here is the corrected version:

  Results: "Khurdopin glacier has a strong build-up signal until the full surge onset. The lower limit of its reservoir area, or dynamic balance line, propagates downward as a surge front during the build-up phase. It is visible as an area of positive elevation change trend slowly propagating down-glacier (Fig. 8.b, area b1). The surge front extends from about 27 km of the glacier head in 2002 to about 33 km in 2015, representing a regular

advance of about 460 m per year, which is approximately 6 times faster than the surface velocity after the front passage, according to the NASA MEaSUREs ITS_LIVE project repository (Gardner et al., 2022). During this period, we do not observe a clear mass transfer from an upper reservoir area , which seems thus different from a slow surge onset"

Discussion: "We now discuss the recent surge of Khurdopin glacier. The propagation of the surge front during build-up has not been observed on this glacier, to our knowledge. The existence of kinematic waves or surge fronts that propagate the surge instability have regularly been observed on other surges (e.g., Cuffey et al., 2010; Kotlyakov et al., 2018; Turrin et al., 2013), with unclear definition of the phenomena. For the Khurdopin glacier, the mechanism seems different from both a kinematic wave or a slow surge onset, with constant thickening after the passage of the surge front during build-up showing the extension of the reservoir area and no upper reservoir area drained. Turrin et al. (2013) observed with velocity data the propagation of a surge front several years before the Bering glacier surge, triggered consecutively to the passage of the activation front down the reservoir area. The activation front for Khurdopin also propagated faster than the surface velocity."

L383: "bulge front"
Just use bulge. Also, as mentioned before, it's not pre-surge.
> The sentence have been changed to "or the propagation of a surge front during the build-up phase preceding the Khurdopin surge" after the main comment of the reviewer.

- L254-255: "representing a regular advance of about 460 m per year, which is approximately 6 times faster than the surface velocity, according to the NASA MEaSUREs ITS_LIVE project repository" I am not sure I get what you mean here - the surface velocity data at Khurdopin clearly shows seasonal behavior with velocities reaching around 400-450 m * yr−1 , starting in 2013 with a quasi-linear increase in velocity up to 2017. Although I might be wrong, i would expect you to be able to see that the surge front advance rate is slower between 2000-2012 than 2012-2017 when the glacier slowly starts to shift to a velocity weakening regime.
> We used annual velocities (much slower) for this estimation, and we did not wish our analysis to go deeper on this topic, although indeed our hovmoller diagram also seems to point toward an acceleration after 2011 at first glance. For the revised version of our manuscript, we will revise this analysis, add details or clarify it.

- L275-76: "The buildup and emptying of the first surge seems weaker than the second one, and extends less up-glacier of the junction, compared to the second surge" Again, refrain from using weaker as it gives the false idea that the surge did not dissipate as much energy - something we have no idea on. The peak velocities of both of Yazghil surges are actually pretty similar and both are visible up to the glacier front in the surface velocity record.
L277-278: "This may be related to the effect of the tributary surge, that stopped at the junction but could have yet increased mass input by a blocking effect.", I really don't get what you mean by that, please explain.
> We will revise our analysis and statements in the revised version of the manuscript. We may remove these sentences if not supported enough.

- This whole section is very confusing. I do not understand what the first sentence is supposed to mean, how can an uncertainty estimate over a quantity reflect the filtering capabilities of a filter ?
> This should be more clear after rephrasing : "The uncertainty estimate of the ALPS-REML algorithm, which is represented in the figures, does not represent the uncertainty of the whole workflow. The performance of the filter is not taken into account in such uncertainty, although it has a major impact on the result"

What does it mean that the surge of Khurdopin shows that "that a discrepancy of a hundred meters is credible on exceptional events." ?

> It means that it is possible to reach an elevation estimate error of 100 m on some dates of a surge, as it is the case of the Khurdopin surge. We give an order of magnitude because it can occur on other glaciers with other values. We clarified the sentence : "The case study of Khurdopin glacier surge shows that a wrong estimate of a hundred meters of our workflow is credible on exceptional events and at precise dates during the surge (Fig. S2.a in Supplement, in 2017)."

In addition, to further test the outlier filtering side of your methodology, you could generate false erroneous measurements and further quantify how well your method performs at filtering simulated outliers.

> We thank the reviewer for their suggestion. It would indeed be interesting, however we do not wish to implement this at this stage, as this may require several days of work to get it done right, and would not change the main results of our study. The way false erroneous measurements is critical to obtain meaningful results, and it would be a whole study in itself.

- L281: "keep true elevations"
Be careful with the use of "true". All measurements are imperfect representations of the "true" elevation, which is by definition, unattainable.

  > This now reads as "to keep accurate elevations observations", and similar replacements have been done in two other sentences.

- L309, Add a full stop before "To test"

  > We divided this part in two paragraph, the second one starting at this sentence.

- L376: "one of the shortest surge cycles in HMA."
Is this from Bhambri et al. (2017) ? Make sure to add proper reference

  > It is a generic information which can be inferred from various inventories, which find shortest cycle durations between 5 to 8 years.The sentence is now:" among the shortest surge cycles in HMA (Bhambri et al., 2017; Sun et al., 2022; Yao et al., 2023; Vale et al., 2021).".

- L378: "Our data suggest it started 1-2 years later, implying a longer quiescence phase of 11-13 years"
Do not make this a general statement on the dynamics of Yazghil glacier - ≈ 8 years of quiescence is not different from ≈ 11 when the number of considered events is 2.

  > It was not intended as a general statement. We precised it to avoid further confusion, "implying a quiescence phase of 11-13 years for this cycle.". Furthermore, Bhambri et al. (2017) are indeed speaking of a cycle duration of 8 years, while we do speak of quiescence duration.

- Figure 1: Readers familiar with surges and HMA will know where the glaciers you mention are, I am not sure this is the case for the broader audience - maybe you could zoom in on a bounding box around the selected glaciers.
I am not sure the whole Kararokam region needs to be displayed since you focus on specific glaciers.

  > The interest of plotting the map of the whole Karakoram region is to show the disparity of DEM observations along stripes, although indeed the area is too large. We will zoom in for the revised version.

- Figure 3
This is a bit of a nitpick here but I would refrain from using two similar colors for the lines in "Interpolation it. 1" and "Interpolation it. 2" - being colorblind, I can't see the difference between them.
In addition, it would be beneficial if you showed the uncertainty associated to each measurement on the plots to the right.

Finally, I see no mention to any Student-T distribution in the paper (because the methods you rely on make no explicit assumption on the distribution of the data). Rename the "t-interval" into "Confidence Interval".

Figure 6
It's really hard for me to see the individual points between raw elevation and filtered measurements.
I think the symbols and the figure in general are just too small.
Again, just remove t-confidence interval and use confidence interval. i think it's too specific for most readers - if they want to know more, they will read Shekhar et al. (2021).

Figure 7
Increase the size of the figure and individual panels.

We will go though a second round of colour blindness test. We will improve these three figures accordingly in a revised version of the manuscript.

- Figure 12
Please add the red A, B and C regions in the captions. It's a shame to have to go into the text to grasp what the figure shows.
We completed the caption with "The areas and trends designated in red are discussed in subsection 5.3. They highlight areas of large surge smoothing or removal (zone A) or overall smoothing of elevation changes (trend B) by the original method (Hugonnet et al., 2021), and artefacts created by the presented workflow (zone C)."

- Table 2
This table is pretty confusing.
I would suggest replacing Table 2 with a figure showing the distributions for each glacier.
This would avoid having 2 columns as the 90th percentile and show the full distribution.
We replaced the table with the following figure :

[Figure]

**Figure 11.** Histograms of the elevation difference between the references DEMs and the DEMs of the workflow interpolated at the same dates. We consider only surge-affected areas. Vertical dotted lines are the median of each histogram. The largest median is 5.18 m (resp. -5.63 m) during surge (resp. during quiescence).

---

## Author Response (AR1)

**Response to the comments from the reviewers**

We thank all the reviewers for their work on our manuscript. Please find below in three different sections (page numbers below) the answer to each reviewer. Our answers are in blue.

| Answer to the comments fr | om William Kochtitzky, reviewer#  | 1page 2 |
|---------------------------|-----------------------------------|---------|
| Answer to the comments fr | om Mingyang Lv, reviewer #2       | page 5  |
| Answer to the comments fr | om Gregoire Guillet, reviewer #3. | page 7  |

We made additional changes to the manuscript and auxiliary files, summarized below:

- We updated the code in the repository to include minor changes. This includes the correction of
  minor errors that does not affect the results presented in this manuscript, but that improves the
  reproducibility to other datasets and areas. Some comments have also been improved.
- We moved the previous appendix to a separate supplement document, with the increase in the number of supplementary figures.
- We simplified the presentation of the general workflow. We replaced the term "pre-filter" (1st submission) by "pre-processing", which is now described in the data section rather than in the methods and is no longer illustrated in Figure 3. Two of the three steps of the pre-processing are very specific to this dataset, and added unnecessary complexity and confusion to the workflow presentation. We modified several figures accordingly (e.g., Fig. 2, 3 and 6 mostly, other figure may only show label update).

**Response to the comments from William Kochtitzky, reviewer #1.**

The authors present a new method of computing elevation changes during surge events when numerous elevation measurements are available for glaciers in a relatively short time period (annual to decadal). The method is novel and advances our knowledge of surging. They provide new insight into several different surge events that have been previously documented. I recommend the paper be published, but I have a few minor comments that could strengthen the paper below. My main criticism is that the authors seem to lack a quantified uncertainty of their results. For example, it would be greatly beneficial to add uncertainty to table 1. I don't think another round of review is necessary, I am happy to see it published after the authors make these minor changes.

**-Will Kochtitzky**

We thank Will Kochtitzky for their careful reading of our manuscript and their comments. We have made all the minor changes suggested, please find below the remaining points addressed. Our answers are in blue.

**Answers to comments**

• Figure 2 – very cool figure, but I am having a hard time understanding it. I think part of the problem is that I don't get the colorbar showing the density of DEMs. I don't see this mentioned in the caption. Is it showing how much of each DEM is good data? Maybe don't make the line and the colorbar blue to add clarity – the caption is confusing which blue you are referring to.

Thanks for the positive feedback. The information given by the 2D histogram / density of DEM was not very relevant and difficult to understand, we have choosen to remove it.

• Page 5 – can you give us a sense of the data that you filtered out? What is the percent of the data that was filtered out of your study?

We added a sentence to the last paragraph of the subsection 4.1 Results / Performance of the outlier filtering: "The pre-processing step removed 46% of the original regional dataset (number of on-glacier pixel), and the filtering step removed a further 42% of the pre-processed dataset (69% removed in total compared to the original dataset)".

• Figure 7 caption – you say in the text what the red circles are, but this should also be added to the caption for clarity:

We completed the caption with the following sentences and added labels for in-text reference: "The red circles (A1-C3) and the dotted lines (a3-5 and d3) show or delimitate areas discussed in the text.".

Line 245-250 – this is very interesting – do you think you are not capturing the mass fully?
 Where could it be coming from? Are parts of the reservoir zone not included in your calculations?

There are indeed parts of the glacier that are not included in the calculation, please find attached a map showing in thick black lines the manually drawn reservoir and receiving areas (Fig. R1.1; for notice all the four glacier delineation in the data repository online, see data availability). However, a similar imbalance is computed when using the full outline (red on the figure): -3081 .106 m³ and 3956 .106 m³ in the reservoir and receiving areas, respectively. We completed a sentence to add this information. The imbalance varies with glaciers and dates chosen (Table 1).

We did not analyse enough surge events in this paper to draw generalities, and we do not have explanations for this. We extended significantly the discussion about crevasses in discussion, subsection 5.2: "The difference between volume gain and loss we estimate is equivalent to a layer of 4.55 ±2.68 m thickness over the surge area. This imbalance is unexpected as the surge occurs over a short time period and mass should be roughly conserved, despite a high uncertainty. The imbalance is quite similar when using two filtered ASTER DEMs over a similar period over this surge, instead of the interpolated series, or when calculated over the full glacier system instead of over the

delineated reservoir/receiving areas. The impact of crevasse opening during the surge on the apparent surface elevation has not been assessed, especially regarding our imbalance, but it may represent a non-negligible volume. The opening of crevasses can be equivalent to up to 0.2 m thickness over regional scale of the Greenland Ice Sheet (Chudley et al., 2025). As inland parts of these regions are largely crevasse-free, we can expect such volume to be significantly larger over the highly crevassed post-surge surface of Hispar glacier, at least one meter magnitude. By mid-2018 our imbalance is equilibrated, as well as is the imbalance of Bhambri et al. (2022) with a end term in 2020 when a number of crevasses have already closed. The Khurdopin and Kyagar glaciers were already crevassed a lot before the surge, and such opening may be less important"

Fig. R1.1.Map of the elevation change over the Hispar 2014-2016 surge and boundaries used for our calculation.

 Table 1 – what are your uncertainties on these measurements? This is critical since any imbalance outside the uncertainty would be a more important signal. Can you get these like you did for figure 6?

> We understand the concern of the reviewer and acknowledge that our manuscript was not very clear regarding uncertainties. Below we detail a number of uncertainties that we clarified in the revised manuscript.

First, to clarify on the previously existing material. On figures 6, 3 and 9, we show confidence intervals that are those of the interpolation algorithms (GP Regression and ALPS) and not of the overall workflow. We do not think that the theoretical uncertainty and confidence interval predicted by the ALPS-REML are credible to assess the results of the overall workflow. For instance, Figure S2.a (in Supplement) shows well that the filter can remove some critical portions of the surge signal (creating data gaps that could also exist in the original dataset) and that is not represented in the interpolation uncertainty. A visual verification of the process permits to detect large biases, but does not give any metric value. This is why we developed the sensitivity analysis, the comparison with external reference DEMs or reference study on a number of events. We added some clarification: « Note that the uncertainty estimate of the ALPS-REML algorithm, which is represented in the figures, does not represent the uncertainty of the whole workflow.» (in preamble of the discussion), and « The confidence interval is valid for the interpolation only and not the whole workflow [...]» (caption of Fig. 6).

- Second, in the revised version we added an estimate of volume transfer uncertainties. We describe the method in the new subsection 3.6. Briefly, we uses a variogram analysis computed by aggregating normalized experimental variances over four surge events and with elevation difference against reference DEMs, and a delineation error. These uncertainties are not informed by the actual correct or erroneous filtering and interpolation of each event, they are thus only indicative. We especially updated the Table 1 with the volume and imbalance uncertainties.
- Figure 10 shouldn't the colorbar be elevation difference? Change implies the difference is real, but if I understand this correctly, the difference should be 0.

This is right and is now corrected.

• Line 330 – are they within uncertainties?

Yes, we added the sentence "It is within uncertainties, although there is more than one year difference between the two estimates periods.". For notice, we cannot retrieve the uncertainty of Bhambri et al. (2022) for the reservoir area estimate, as we cannot retrieve the uncertainty they allocate to the Khani Basa tributary.

• Line 338 – the main sentence on this line is grammatically incorrect, I am not sure what you are trying to say

This now reads as "The existence of kinematic waves or surge fronts that propagate the surge instability have regularly been observed on other surges" (previously, "kinematic wave propagating the surge front").

• Figure 12 – need to add what areas A, B, and C mean to the caption

We completed the caption with "The areas and trends designated in red are discussed in subsection 5.3. They highlight areas of large surge smoothing or removal (zone A) or overall smoothing of elevation changes (trend B) by the original method (Hugonnet et al., 2021), and artefacts created by the presented workflow (zone C)."

**Response to the comments from Mingyang Lv, reviewer #2.**

This study provides a new quantifying approach to describe the elevation change patterns of glacier surge events, based on high temporal resolution successive DEMs. The method applied to automatically detect the surge elevation signals is inspiring, and its application on large scales would be quite interesting. I recommend the paper be published after the following suggestions are considered by the authors.

We thank Mingyang Lv for their review and feedbacks on our manuscript. We have made all the minor changes suggested, please find below the remaining points addressed. Our answers are in blue.

**Answers to comments**

About the selected time series

To better illustrate the elevation change patterns over the reservoir and receiving areas during surge events, I suggest the authors to add more number of typical time series analysis results in the paper body or Appendix.

We added in supplementary file 6 to 9 time series for each of the analysed glacier, sampled at regular interval (each 5 or 2.5 km) along the selected centerlines.

About the method

Start date and end date of glacier surge events are one of key information to describe the surge process. It will be more convincing if the authors add more information about the criterion to identify the start and end dates of these glacier surge events, based on the processed DEM time series datasets.

The identification is visual, looking for abrupt changes of the elevation time series, we do not propose a measurable criterion in our method. This is now explicitly written in the text. We added these sentences to the subsection 3.4 Methods / Volume transfer estimate: "It is difficult to constrain precisely the initiation and termination of surges. The surge dates (Table 1) are estimated visually from two sources: the pre-filtered timeseries and the interpolated elevation changes. None of these sources permits to be sure of the exact month of start or end of the surge. We estimate the dates from interpolated elevation change (e.g. Fig. 8) when computing volume transfers, such "apparent" dates are less exact but capture the overall mass transferred in our dataset generated. We may also estimate the dates from pre-filtered time series (not affected by filtering and interpolation defects) for information or validation, which permit to be more exact although we are still limited by the number of observations. For example, for the time series in Supplement Fig. S2.a (from the Khurdopin glacier): the surge period estimate at this location from the interpolated time series would be around 2016-06 to 2019-02, and 2016-12 to probably around 2017-10 using trends (but between 2017-06 and 2018-07 due to observations, time series at other locations are required).".

About the temporal information of the glacier surge events
 In the Section 4.3, the authors used multiple temporal expression formats simultaneously, such as "mid-2014" in Line 234 and "late 2015" in Line 264. According to Fig. 8 and Tab. 1, the start and end dates of surge events are described in monthly units. In order to show the method could realize monthly glacier surge monitoring, the temporal expression formats could better be unified as monthly units.

As discussed in the previous comment, our approach for dating surges is subjective, and our final dataset is not precise up to the month in a number of situations (artefact, smoothing, diverging trend depending on the glacier's location...). Even with ideal data and for a fast-trigger surge, defining a strict month may not be possible or relevant. Therefore, we use precise dating only for volume transfer calculation, when it is necessary to choose a specific date.

• Figure 2 – in this figure, the orange part is easy to understand, but the blue part is not. The legend of this figure could be improved.

The figure has been simplified (following the comment of reviewer #1), and the caption should now be clearer.

• Line 254 – it seems like "glacier source" is not a common glaciology terminology, perhaps what you are referring to is "the highest peak of the glacier centerline" or "the top end (head) of the glacier". Please check it. (also Figure 8)

In agreement with the comment of another reviewer, we have chosen to go with "glacier head".

• Table 1 - you may also add the areas of reservoir and receiving polygons for each glacier. We added the area of each zone in the table. To make space for uncertainty estimates, we removed the column of data gap proportion, which is now given in the caption: "For these glaciers, the percentage of data gap after the workflow presented in this study is ranging from 0 to 5.6% (median of 1.4%), and after bilinear interpolation it is of 0 to 0.8% (median of 0.2%)".

| Glacier
RGI 7.0 code | Date start
[time series] | Date end
[time series] | Reservoir
vol. change
[Surface area]                  | Receiving
vol. change
[Surface area]                | Imbalance                                                |
|-------------------------|-----------------------------|---------------------------|-------------------------------------------------------------|-----------------------------------------------------------|----------------------------------------------------------|
| Hispar                  | 2014-01                     | 2016-09                   | -2411 ±373 x 10 6 m 3                 | $3110 \pm 175 \times 10^6 \text{ m}^3$                    | 700 ±413 x 10 6 m 3                |
| 21670                   | [2014-05]                   | [2016-06]                 | [106 km 2 ]                                      | [48 km 2 ]                                     | 4.55 ±2.68 m                                             |
| Yazghil                 | 2003-07                     | 2007-01                   | $-32 \pm 30 \times 10^6 \text{ m}^3$                        | $63 \pm 26 \times 10^6 \text{ m}^3$                       | -31 ±40 x 10 6 m 3                 |
| 21865                   | [2004-01]                   | [2006-08]                 | [8 km 2 ]                                        | [6 km 2 ]                                      | -2.19 ±2.80 m                                            |
| Khurdopin               | 2016-03                     | 2019-03                   | -801 ±134 x 10 6 m 3                  | 711 ±63 x 10 6 m 3                  | $-90 \pm 148 \times 10^6 \text{ m}^3$                    |
| 14958                   | [2016-04]                   | [2017-07]                 | [33 km 2 ]                                       | [15 km 2 ]                                     | -1.9 ±3.12 m                                             |
| Kyagar                  | 2012-11                     | 2017-01                   | -271 ±84 x 10 6 m 3                   | 267 ±37 x 10 6 m 3                  | -4 ±92 x 10 6 m 3                  |
| 14958                   | [2013-10]                   | [2015-12]                 | [21 km 2 ]                                       | $[8 \text{ km}^2]$                                        | -0.12 ±2.75 m                                            |
| Kyagar without artefact | _                           | =                         | $-228 \pm 97 \times 10^6 \text{ m}^3$ [20 km 2 ] | $267 \pm 37 \times 10^6 \text{ m}^3$ [8 km 2 ] | 39 ±104 x 10 6 m 3
1.33 ±3.54 m |

Table 1 of the manuscript, with the new field « Surface area ».

- Table 2 the Product ID and Acquisition Time of reference DEMs may be added to Appendix.
   We added a corresponding table in appendix, and reference it at the beginning of the 5.1 subsection: "We use SPOT5 HRS and SPOT6 DEMs generated by Berthier and Brun (2019), and along-track HMA DEMs (Shean, 2017) ( (list in Table S2 of the Supplement)).".
- Figure 10 the labels of points TSa-c should be indicated in both Fig. 6 and Fig. 10.

  We added the labels in Fig. 10. We did not changed Fig. 6, as the labels (a-c) are already the same as TS(a-c), we did not write exactly TS\*\* for conciseness and clarity.

**Response to the comments from Gregoire Guillet, reviewer #3.**

The manuscript proposed by Beraud et al. presents a new methodology to filter outliers and estimate glacier surface elevation from dense digital elevation model time series.

Their study relies on the same data as the Hugonnet et al (2021) study.

Instead of Gaussian Process Regression, Beraud et al implement an additional LOWESS filtering step, before feeding the fitlered time series to a localized b-splines scheme in order to interpolate surface elevation measurements.

While the work and the methods shows promise, I strongly recommend that the authors take the necessary steps to strengthen the manuscript.

The paper appears to be in an early stage of development and would benefit from significant revisions before it is considered for publication: sections 4 and 5 are particularly challenging to understand. With the necessary improvements, I am confident the paper has the potential to make a valuable contribution to the field.

I recognize, from first hand experience, the amount of effort that went into this work. This is why I want to restate my strong support for the manuscript as I think it brings an important contribution to the field. I strongly encourage the authors to address the points I have highlighted, as doing so will enhance the clarity, rigor, and overall impact of the work.

**Greg Guillet, University of Oslo**

We thank Gregoire Guillet for his careful reading, his comments and help in improving our manuscript. We considered all the comments. The minor ones have been accepted and do not appear in this response document. Please find below our answer and larger changes in response to other comments. Our answers are in blue.

**Answers to general comments**

 From this version of the manuscript it is not clear at all to me how uncertainties are calculated, nor what do they actually represent. More details on this throughout the manuscript are severely needed.

We understand the concern of the reviewer and acknowledge that our manuscript was not very clear regarding uncertainties. Below we detail a number of uncertainties that we clarified in the revised manuscript.

- First, to clarify on the previously existing material. On figures 6, 3 and 9, we show confidence intervals that are those of the interpolation algorithms (GP Regression and ALPS) and not of the overall workflow. We do not think that the theoretical uncertainty and confidence interval predicted by the ALPS-REML are credible to assess the results of the overall workflow. For instance, Figure S2.a (in Supplement) shows well that the filter can remove some critical portions of the surge signal (creating data gaps that could also exist in the original dataset) and that is not represented in the interpolation uncertainty. A visual verification of the process permits to detect large biases, but does not give any metric value. This is why we developed the sensitivity analysis, the comparison with external reference DEMs or reference study on a number of events. We added some clarification: « Note that the uncertainty estimate of the ALPS-REML algorithm, which is represented in the figures, does not represent the uncertainty of the whole workflow.» (in preamble of the discussion), and « The confidence interval is valid for the interpolation only and not the whole workflow [...]» (caption of Fig. 6).
- Second, in the revised version we added an estimate of volume transfer uncertainties. We describe the method in the new subsection 3.6. Briefly, we uses a variogram analysis computed by aggregating normalized experimental variances over four surge events and with elevation difference against reference DEMs, and a delineation error. These uncertainties are not informed by the actual correct or erroneous filtering and interpolation of each event, they are thus only

indicative. We especially updated the Table 1 with the volume and imbalance uncertainties.

L360-370:

Be a bit more specific. Please add uncertainty estimates. Mention that differences you show are between median values

L385: "The order of magnitude of the imbalances corresponds to the order of magnitude of the measurement uncertainty"

Can't agree more! Please add them.

Tables: Please add uncertainty estimates in all tables - as a hunch feeling I would typically assume that this is what drive the reported imbalance (except for Hispar).

Our answer to the general comment covers these three minor comments above.

Section 4 needs a vocabulary overhaul. The authors use very unspecific language which makes
it harder to grasp what is the point they are trying to make. I have addressed some of these in
my specific comments.

As it stands, the writing in Section 5 makes it very hard to understand. I had to re-read some sentences multiple time to make sure understood the statements correctly. I think significant efforts are needed to

A lot of the sentences start with unspecific pronouns such as "this" or "it" - which required me to backtrack a few times. This severely hampers the readability of the manuscript.

Acronyms are not always defined on their first use - please check this.

Finally, there are quite a lot of frenchisms and typically french sentence constructions - it's not a major problem, but the manuscript would gain in readability if the authors addressed this.

The overall article and this sections particularly have been proofread and improved. The specific comments are addressed in this answer.

There are some problematic statements which show some confusion between GPs and splines. I think these can and should be addressed with minimal changes by removing the statements I have highlighted in my specific comments. In addition, the authors need to reframe comparisons between methods in the manuscript as such: comparisons between the method proposed by the authors and the GP from Hugonnet et al. (2021).

Regarding the confusion between GPs and splines, we answer directly to the specific comment below in this document.

About clarifying the different comparisons: we completed the end of the introduction with "We evaluate the performance of the workflow compared to the results of Hugonnet et al. (2021). We also compare the surge characteristics such as volumes transferred to other products and studies [N.B.: studies of the literature, that worked on the case studies of a few surge events]." We also already insist on the fact that we compare a lot our workflow to the GP regressions as implemented by Hugonnet et al. (2021). Part of the new subsection 3.4 Gaussian Process regression states that our comparison does not apply to all possible settings of GPR.

• We have revised the writing of the article: proofreading of unspecific pronouns, acronyms, frenchisms...

**Answers to specific comments**

"We compare the produced dataset to previous studies [...]".
 If I am not mistaken, you only compared it to the Hugonnet et al. (2021) results. I would hence keep it singular here.

With this sentence, we also refer to the work done to compare the four surge events we analyse with the dataset produced. In the subsection 5.2 Discussion / Comparison of surge characteristics with the literature, we compare dates, volume transferred etc. against other studies (e.g., Bhambri et al. (2022), Steiner et al. (2018), Gao et al. (2024)...).

I generally agree with all the statements made in the introduction and they are accurate.

However, as it stands, I find the structure of this introductory section quite confusing. The authors often switch between very broad statements to nichely precise descriptions of methods/problems without clear guidance of the reader into why this matter, which ultimately leads to significant repetition.

This is especially true between lines 30 and 50, which is followed by a lengthy statement on glacier surface velocity inversion methods which seems a bit out of place - I would either make the point clearer as to why this is important here or remove it.

Within these lines, I suggest a little restructuring:

- 1. What are surges and why are they important (essentially your lines 16 to 23)
- 2. Why are elevation changes particularly important when studying surges (which is basically your lines 35 to 34) but I would focus on 1) visual interpretation (i.e. mostly inventory efforts) and 2) quantitative interpretation i.e. computation of shear stress, and even more, ice thickness/bed geometry inversions. I would remove the mentions to velocity products
- 3. Problems with the data and then the problems with the methods used up to know and how you're solving them.
  - We reorganised the introduction, with a few minor changes of content.
- Paragraph starting at line 52 May I suggest adding my own work Guillet and Bolch, 2023 where we develop a Bayesian outlier filtering and uncertainty quantification framework to compute thickness changes from DEMs, specifically for surge-type contexts.

Thanks for this suggestion. We included this work by these few lines: "A recent study has exploited a Bayesian framework by inference applied to elevation change to filter outliers, which requires prior knowledge from diverse sources (Guillet and Bolch, 2023). It has been tested on surge-type glaciers, and it applies equally to dense time series."

 Paragraph starting at line 68, I would suggest to bolster this paragraph and be a bit less succinct - you have done a lot of work here and this is a good place to showcase a short summary.

We completed 3 sentences to add a few details. This now reads as "In this study, we aim at developing a workflow to analyse outlier-prone, moderate-precision and high-temporal-resolution elevation dataset adapted to the specificity of surge events. We use established algorithms to filter outliers and interpolate elevations at monthly scale while preserving surge elevation signals. We apply it to an ASTER DEM dataset from Hugonnet et al. (2021). We produce a regional dataset in the Karakoram region covering more than 100 surge-type glaciers. We evaluate the performance of the workflow compared to the results of Hugonnet et al. (2021). We also compare the surge characteristics such as volumes transferred to other products and studies."

- L92: The temporal sampling is heterogeneous in time and space. Rephrase.
   This now reads as "The temporal sampling is not regular in time, and parts of the mountain range have about twice less DEMs than others".
- It took me 2 reads of these sentences to understand what was clearly meant. First, "below 50 days apart" should be "less than 50 days apart", but a more general question is whether both sentences are needed. I feel they give redundant information, that is illustrated in Figure 2. I would rephrase to avoid this repetition and clarify the statement.

We removed the first sentence.

• Is this 400m threshold arbitrary? It's not a problem if it is - I am wondering if there is a reference to back this up or if its from the author's personal experience with the data.

Also, the notion of "spatial" filter is a bit confusing as you are filtering pixels that show and absolute difference in elevation (z coordinate) between the ASTER and GLO-90 DEMs, correct? Or are you operating in the x, y plane?

Yes the 400 m threshold is arbitrary. Some surges with up to 300 m elevation change have been observed (e.g. Chamshing Glacier II, in our dataset) and we do not know any observation with more than this value, our threshold is thus quite conservative. On two

glaciers of the Karakoram, Guo et al. (2020) and Bhambri et al. (2020) observed thickenings up to 180-190 m.

The name is indeed not appropriate, thanks, "absolute filter" would be clearer. However, the revised manuscript do not name it anymore. We simplified the presentation of the general workflow. We replaced the term "pre-filter" (1st submission, where this absolute filter was presented) to "pre-processing", and we do not present it in the workflow section but in the data section and more briefly. Two of the three steps of the pre-processing are very dependant to this dataset, and added unnecessary complexity and confusion to the workflow presentation. We modified several figures accordingly (e.g., Fig. 2, 3 and 6 mostly, other figure may only show label update).

This will probably reveal my total lack of knowledge on DEM generation methods but, is the
correlation score a sufficient enough metric to discard data?
 Or is there value in computing a median DEM from all the available DEMs for this day and using
this instead?

First of all, we did remove mentioning the steps of removal of pixels with 51% correlation score and of strip merging. Indeed, these steps may be confusing for the reader while it is not the focus of the paper, and they are very specific to our dataset. The answer to the point above explains in more detail how we modified the presentation of the workflow.

The reviewer is correct that the correlation score is not always the best metric to exclude some data, we write in the subsection 4.1 Results / Performance of the outlier filtering: "This filter oversensitivity occurs on time series with scattered elevations, and it is often due to scattered elevations and to the correlation score that is not very representative of the actual pixel quality: outliers may have lower uncertainties than more accurate observations (e.g., Fig. S2.e or S7 at 15 km in Supplement)." Here below is the figure (Fig. R3.1), showing some points of various elevation error (representative of the correlation scores for a pixel time series) mixed altogether:

Fig. R3.1. LOWESS workflow over a sample time series.

About merging stripes based on the maximum correlation score versus a median of values, yes both are possible. A few specific cases where in favour of using correlation score. However, overlaps represents a small portion of the data, so the impact should be very limited.

• L111-117 This could be condensed a bit.

Filtered

We removed 2 sentences out of these 6.

 L 125-126: "For our dataset, the output of the regression is to sensitive to noise overall and too smooth over surges to be used directly as an interpolation of the elevation, so we use it for filtering only."

This is a pretty important point which I feel is a bit brushed over by the authors here, as it is pretty well documented that LOWESS struggles in non-stationary contexts.

Thank you for your remark. We slightly disagree with the fact that LOWESS is not well suited for non-stationary contexts. Indeed,the LOWESS regression is a "local" regression, which uses a local kernel to mitigate non-stationarity over the full time series

and adapt its fit locally. But it is true that the smoothness of the LOWESS is set globally for the whole time series and it may fail at capturing very abrupt changes. We apologize, but we did not understand what kind of changes were expected here and we did not find good references illustrating the limits of LOWESS in non-stationary contexts, therefore we did not make any change to the text. But we would welcome more specific suggestions if changes are needed.

L165-170. I have to say I strongly disagree with this statement and I think there is a bit of confusion that needs to be addressed.

While GPs do require a prior on the functional form to fit a given dataset, the shape of the kernel function can actually be interpreted as physical variations in the latent variable, i.e. the shape of the Linear + Periodic kernel used in Hugonnet et al (2021) reflects the sub-linear and periodic signal one could expect from surface elevation changes.

Framed as a causal inference problem, a GP should not be used with prior belief on the structure of the data, but on the physical process the data represents.

ALPS also puts a "prior" on the functional form to fit, as it relies on a combination of b-splines to adequately approximate local regions of the data. Even if you were interpolating with linear functions, you would be making an assumption that there is no oscillatory behavior between data points - which, by definition is a prior assumption.

The main difference here is thus that the "prior-like" assumptions are expressed through, for example, the smoothness of the derivative (splines) instead of a covariance function (GP). In a more general way, any model requires prior assumptions.

As a side note, I want to point out that standard smoothing splines represent a special case of GPs, as shown by Kimeldorf and Wahba (1970).

In a highly-abstract way, you are, implicitly, fitting different GP models to different regions of the dataset - one should not stretch this analogy too far, as it breaks down when considering uncertainty estimates, as GPs are probabilistic.

All in all, I would argue against the generalization of this statement to every GP model, and only focus on the one defined by Hugonnet et al. (2021), otherwise this is not an apple-to-apple comparison.

You are comparing the results of two different methodologies and how they fare at interpolating time series of surface elevation data in the presence of transient physical events, no more, no less. In addition, and I think this point is an important plus for the author's approach: I would imagine ALPS is more scalable than any GP model over larger sample and dimensions datasets, as GP are knowingly computationally expensive.

This is a strong pro for your method and I think you should address a bit more, as surface elevation time series are likely to get denser in the future.

Kimeldorf, George S., and Grace Wahba. "A Correspondence Between Bayesian Estimation on Stochastic Processes and Smoothing by Splines." The Annals of Mathematical Statistics 41, no. 2 (1970): 495–502. http://www.istor.org/stable/2239347.

We thank the reviewer for their explanations, and acknowledge that our text was not correct about these methods. My earlier view on B-splines was simplistic with a "user" point of view and I was lacking the theoretical background. We created a specific subsection (3.4) for GPR. Although we keep a very succinct level compared to reviewer's explanations, the modified paragraph now reads as:

"Gaussian Process (GP) regression, equivalent to kriging, is a non-parametric method that relies on estimating the data covariance to provide an optimized interpolator (Cressie, 1993; Rasmussen and Williams, 2005; Williams, 2007). Under certain assumptions, including

notably second-order stationarity, GP regression has been shown to be the "best linear unbiased predictor". It is the method used by Hugonnet et al. (2021) on this same dataset, to compute long-term mass balance estimations worldwide. We use a GP covariance with terms estimated in Hugonnet et al. (2021) through a global variogram analysis. This analysis identified several kernel components (periodic, local, linear...), that are not specifically tuned for surges.

We note that, contrary to GP regression, ALPS approximates the data with polynomials under the assumption of a degree of smoothness of the data, with no need for us to inform the behaviour of the data. Although both GP regression and ALPS165 need domain knowledge to decide the covariance kernel and spline degree/penalty respectively, from a user's perspective using GPs can be more complex owing to the well studied difficulty of optimizing the kernel, mean function and dimensionality (Pu, 2024). For ALPS on the other hand, we simply manually select degrees and penalty orders from a small set of choices."

 L188-189: "Modifications of this kernel to allow for stronger changes in elevation have not proven to be efficient enough"
 I would be a bit more careful and specific here, as I know of successful attempts at this with different GP kernels.

This now more developped in the new subsection 3.4 Gaussian Process regression: "Reparametrization of the kernel used by Hugonnet et al. (2021) gave slightly worse results than those obtained with the ALPS-REML method. Our limitation with GP regression lies in the kernel definition which is done according to the variance170 of elevation changes. Each surge event is different in variances, which is also very different from the data variance in quiescent periods or on non-surge-type glaciers. We tried different settings of the kernels, that differs from the study of Hugonnet et al. (2021). We removed the seasonal term of the model. The length scale and the magnitude parameters of the remaining terms were manually tuned after testing. We added radial basis function terms of length scales of few months and with a variance of a few tens/hundreds of square meters. The kernels that provided a suitable interpolation were slightly outperformed by the ALPS-REML algorithm. This could be reevaluated for other datasets (for e.g. less noisy), more complex or adapted GP regression processes and future advances (e.g., de-trending before GP regression or using other predictors)."

- L 190-192: "It does conserve nearly all known surge events in our study area and period, with one exception being surge events with strong melt before and after the surge"
   When you say study area and period, you mean the 4 surges you look at, correct? [...]
   It also applies to a broad number of surge events in the Karakoram, although we did not check manually each surge-type glacier. We rephrased our statement this way: "It conserves well the surge signal of 3 out of the 4 events we analyse in subsection 4.3, and this observation seems to extend to a number of surge events in Karakoram. One exception is periods of low temporal density during surge events, especially when combined to strong melt before and after the surge."
- L197-203 This whole section is confusing. [...]
   What I get from this section is that your method works better than Hugonnet et al (2021) onglacier, in parts with that are relatively smooth, but tend to over-filter in areas of low contrast/rough terrain am I correct? In any case please make the section easier to understand.

We applied the minor changes suggested and we simplified some parts of the subsection. We added the following sentence at the beginning of the last paragraph: "To summarize, our filter permits to preserve better the surge signals that were filtered out in the workflow of Hugonnet et al. (2021). However, the new filter is more noise-sensitive over textureless accumulation areas and rough terrain, leading to data gaps or artifacts with large elevation changes."

L205 Again, this is confusing. Please rephrase.

This sentence and the one at the end of the section 2 "Data" have been thought through quite much during the writing process already, without any simple solution. We could reason by probability, which we did not keep: "Any random date in the time series period have 40% (75%, respectively) probability to fall between unfiltered observations less than a year (two years, respectively)". You can give your thoughts on this on a second

round of review.

As a reminder, the current version is "After filtering, nearly 30% (62%, respectively) of any date in the time series periods are between observations less than 9 months apart (one and a half years, respectively)". For notice, values have changed due to the removal of a "pre-filter" step in the workflow (see more details in the answer about the 400m-threshold).

• L209-212; An interesting point is that the GP used in Hugonnet et al (2021) shows an increasing trend in surface elevation, completely omitting the actual data. GPs tend to "fall-back" to the median when there is no data but here, both the median and the uncertainty increase. Can you plot the uncertainty of each measurement?

We show on the figure below (Fig. R3.2) the time series coloured by the elevation error (in m), which is estimated with the workflow from Hugonnet et al. (2021). We added the sentence "It is noteworthy to mention that by its design, the original kernel is optimized to preserve a linear trend to extrapolate out of the observation period of each pixel."

Fig. R3.2. Time series with the interpolation of Hugonnet et al. (2021) and the per-pixel elevation error estimates.

• L215: "[...] creating wavelet artifacts", The term wavelet design something different in signal processing and I would refrain from using it here. I would use "spurious high frequency

oscillations" or something similar.

We thank the reviewer for the appropriate suggestion, we replaced our old term with it.

- L218: have weakest changes, Weaker. But please consider changing to a more specific term.
   We changed to "show smaller elevation changes"
- L220-221: "Some glaciers are more affected by data gaps than others, in agreement with areas with a low number of observations (Fig. 1, e.g. Shisper glacier)", Shisper is not highlighted on Figure 1 and is not in the studied glaciers.

Showing Shisper glacier was not intended, it was just an additional piece of information because our dataset did not allow us to add its surge to our panel due to large data gaps, even though it is a well-studied case. For clarity, we removed the mention of Shisper glacier.

• L227: "with a decreasing speed (2009-2012, a1)", This figure does not show the velocity. Also, the reference to the figure is broken.

We here talk of the surge front propagation, which is visible on the Hovmoller diagram, we do not talk about ice velocity. To avoid further confusion, we replaced "speed" by "rate of propagation". The reference to the figure has been corrected.

• L246: Add uncertainty estimates - at least some part of the discrepancy is in there. I imagine there is an underestimation in the surface area of the reservoir zone? Did you account that the surge also simultaneously drains the northern (Yutmaru?) tributary (centerline RGI2000-v7.0-L-14- 27499)?

As discussed previously, we did not provided uncertainty on interpolated elevations on purpose (and thus on volume transfer estimates), but we provide some in this revised version (see the first answer to general comments).

The surface imbalance with the uncertainty estimates is of 4.55 ±2.68 m. Indeed some parts of the discrepancy should lies in the area delineations, although the choice of dates could have an even greater impact (we state that "By mid-2018 [end term of the DEM difference] our imbalance is equilibrated").

Yes we include the northern/Yutmaru tributary. Here below (Fig. R3.3) we show the elevation map with the distinct reservoir and receiving areas (in black) we used for calculating the surge volume (for notice, delineations are also provided in the public repository, see data availability). We have a similar imbalance when we compute it at the scale of the glacier system (the whole red outline, including Kunyang tributary).

Fig. R3.3. Map of the elevation change over the Hispar 2014-2016 surge and boundaries used for our calculation of transferred volume.

• L253: "The "build-up front" or kinematic wave"; To stay consistent with current terminology I would use surge front, not build-up front or kinematic wave.

L337: "pre-surge thickening front or kinematic wave"; Same as before. The propagation of a thickening front is one of the definitions of a surge. In your case it is still the early stage and has not reached the dramatic proportions it will eventually attain. Stop using kinematic wave.

Thanks for raising this point. It is indeed different from a kinematic wave, although for example Turrin et al. (2013) use both terms ambiguously in a similar situation. We removed the term "kinematic wave". However, we kindly disagree with the reviewer about the surge phase. We do not interpret the build-up that happens from 2001 until 2016 as the active phase, which would involve the thinning of the reservoir zone and thickening of the receiving zone, but as the quiescence phase. According to Cuffey & Paterson (2010, section 12.1), "During the quiescent period, converging ice flow thickens the upper reservoir area and the lower part stagnates and thins by ablation." (and similar definition in the glossary of Cogley et al. (2011)) which is exactly what we observe during this period on figure 8b. We interpret the elevation-change pattern as a geometric readjustment of the glacier following the previous surge, which occurred 2-3 years before our first observation (Quincey et al., 2011). However, we agree that the text and terminology was unclear and we modified extensively several paragraphs of this section for the sake of clarification and revised the terminology. Here is the corrected version:

Results: "Khurdopin glacier has a strong mid-glacier thickening signal until the full surge onset. The distinct area of positive elevation-change trend extends down-glacier during at least 15 years (Fig. 8.b, area b1). This mass build-up may be the geometry readjustment of the glacier in its quiescent phase, after the previous surge in 1998 (Quincey et al., 2011). The lower limit of this build-up area propagates downward from about 25.5 km of the glacier head in 2001 to about 33.5 km in 2015. The limit advances approximately 600 m per year during this period, which is about 7 times faster than the surface velocity (measured 2 km upstream of the front), according to velocities (temporal baseline from 300 to 430 days) from the NASA MEaSUREs ITS\_LIVE project repository (Gardner et al., 2022). During this period, we do not observe a clear mass transfer from an upper reservoir area, which thus seems different from a slow surge onset."

Discussion: "We now discuss the recent surge of Khurdopin glacier. The geometry readjustment and the propagation of a build-up front during quiescence has not been

observed on this glacier, to our knowledge. The existence of kinematic waves or surge fronts that propagate the surge instability have regularly been observed on other surges (e.g., Cuffey and Paterson, 2010; Kotlyakov et al., 2018; Turrin et al., 2013), with unclear definition of the phenomena. For Khurdopin glacier, the mechanism seems different from both a kinematic wave or a slow surge onset. As opposed to these processes, here we observe a constant thickening after the downward extension of the build-up area with no upper reservoir area drained. Turrin et al. (2013) observed, with velocity data, the propagation of a surge front (moving as a kinematic wave) several years before the surge of Bering glacier, triggered by the passing of the front through the reservoir area. The build-up lower limit for Khurdopin also propagated faster than the surface velocity."

L383: "bulge front"

Just use bulge. Also, as mentioned before, it's not pre-surge.

We removed the term "bulge front" during our reformulation.

• L254-255: "representing a regular advance of about 460 m per year, which is approximately 6 times faster than the surface velocity, according to the NASA MEaSUREs ITS\_LIVE project repository" I am not sure I get what you mean here - the surface velocity data at Khurdopin clearly shows seasonal behavior with velocities reaching around 400-450 m ★ yr−1, starting in 2013 with a quasi-linear increase in velocity up to 2017. Although I might be wrong, i would expect you to be able to see that the surge front advance rate is slower between 2000-2012 than 2012-2017 when the glacier slowly starts to shift to a velocity weakening regime.

We use annual surface velocities for this estimation (much slower than 400-450 m/year). Our elevation dataset does not permit to capture the seasonal signal of the advance, and we compute the advance from long term trend. Indeed our hovmoller diagram also show an acceleration after 2011 at first glance, or more likely 2013. Fig. R3.4 below compare the surface velocity and the elevation change trend. From the frontal position (red drawing) estimated from elevation change trend, we get a median velocity of the front propagation of 600 m/year (standard deviation of 344 m/year with this certainly too much precise drawing). At the front location and timing, it is associated with a median smoothed ice surface velocity of 31 m/year (std=7 m/year). With a time lag of 2 years, it is 53 m/year (std=8 m/year). 2 km before the wave front and no time lag, the mean surface velocity is 83 m/year (std=9 m/year).

With this more accurate estimate, we updated the text:

"The lower limit of this build-up area propagates downward from about 25.5 km of the glacier head in 2001 to about 33.5 km in 2015. The limit advances approximately 600 m per year during this period, which is about 7 times faster than the surface velocity (measured 2 km upstream of the front), according to velocities (temporal baseline from 300 to 430 days) from the NASA MEaSUREs ITS\_LIVE project repository (Gardner et al., 2022)."

Fig. R3.4: surface velocity (from ITS\_LIVE) and elevation change trend (workflow presented) sampled along the centerline of the Khurdopin glacier. The velocity is the rolling mean of velocity pairs with temporal baselines from to 300 to 430 days, to avoid seasonal patterns. The orange contour is the 50 m/year velocity threshold. Black contours are absolute 4 m/year elevation change trends. The red drawing is the front position estimated from the elevation change trend.

• L275-76: "The buildup and emptying of the first surge seems weaker than the second one, and extends less up-glacier of the junction, compared to the second surge" Again, refrain from using weaker as it gives the false idea that the surge did not dissipate as much energy - something we have no idea on. The peak velocities of both of Yazghil surges are actually pretty similar and both are visible up to the glacier front in the surface velocity record.

L277-278: "This may be related to the effect of the tributary surge, that stopped at the junction but could have yet increased mass input by a blocking effect.", I really don't get what you mean by that, please explain.

More careful checking of time series and velocity fields shows that the reservoir area may not extend farther than the first one. The concerned area seems subject to more complex dynamics not in the scope of this paper. We removed both the comparison between the two surges and the argument of the blocking effect. We added the mention of some seasonal elevation pattern certainly fitted.

The paragraph in the subsection 4.3 now reads as:

"Our dataset captures a full surge cycle of Yazghil glacier. On this glacier, the surge signal has a low amplitude (approximately ten metres) compared to the time series, and thus noise is often overfitted resulting in frequent interpolation artefacts. Some seasonal signal seems also to be fitted, for example during the period 2013-2016 thanks to denser and consistent time series (horizontal lines on Fig. 8.d). A surge starts around august-to-november 2003 and ends around october 2016 to february 2007 (Fig. 8.d, area d1), and a new surge starts in 2016 or 2017 (the end is not captured; 8.d area d2). The build-up phase of the second surge is visible, representing about half of the quiescence phase (Fig. 8.d area d3, delimited by dotted lines d3 on Fig. 7.d). One of the tributaries of Yazghil glacier (junction at km 18) is also surge-type, and seems to have surged during our study period in about 2008-2013."

 This whole section is very confusing. I do not understand what the first sentence is supposed to mean, how can an uncertainty estimate over a quantity reflect the filtering capabilities of a filter?

This should be more clear after rephrasing: "Note that the uncertainty estimate of the ALPS-REML algorithm, which is represented in the figures, does not represent the uncertainty of the whole workflow."

What does it mean that the surge of Khurdopin shows that "that a discrepancy of a hundred meters is credible on exceptional events." ?

It means that it is possible to reach an elevation estimate error of 100 m on some dates of a surge, as it is the case of the Khurdopin surge. We give an order of magnitude because it can occur on other glaciers with other values. We clarified the sentence: "The case study of Khurdopin glacier surge shows that a wrong estimate of a hundred meters of our workflow is credible on exceptional events and at precise dates during the surge (Fig. S2.a in Supplement, in 2017)."

In addition, to further test the outlier filtering side of your methodology, you could generate false erroneous measurements and further quantify how well your method performs at filtering simulated outliers.

We thank the reviewer for their suggestion. It would indeed be interesting, however we do not wish to implement this at this stage, as this may require a certain amount of time to get it done right, and would not change the main results of our study. The way to generate false erroneous measurements is critical to obtain meaningful results.

L281: "keep true elevations"

Be careful with the use of "true". All measurements are imperfect representations of the "true" elevation, which is by definition, unattainable.

This now reads as "to keep accurate elevations observations", and similar replacements have been done in two other sentences.

L309, Add a full stop before "To test"

We divided this part in two paragraph, the second one starting at this sentence.

L376: "one of the shortest surge cycles in HMA."

Is this from Bhambri et al. (2017)? Make sure to add proper reference

It is a generic information which can be inferred from various inventories, which find shortest cycle durations between 5 to 8 years. The sentence is now: "among the shortest surge cycles in HMA (Bhambri et al., 2017; Sun et al., 2022; Yao et al., 2023; Vale et al., 2021)."

L378: "Our data suggest it started 1-2 years later, implying a longer quiescence phase of 11-13 years"

Do not make this a general statement on the dynamics of Yazghil glacier -  $\approx$  8 years of quiescence is not different from  $\approx$  11 when the number of considered events is 2.

It was not intended as a general statement. We precised it to avoid further confusion, "implying a quiescence phase of 11-13 years for this cycle.". Furthermore, Bhambri et al. (2017) are indeed speaking of a cycle duration of 8 years, while we do speak of quiescence duration.

Figure 1: Readers familiar with surges and HMA will know where the glaciers you mention are, I
am not sure this is the case for the broader audience - maybe you could zoom in on a bounding
box around the selected glaciers.

I am not sure the whole Kararokam region needs to be displayed since you focus on specific glaciers.

The interest of plotting the map of the whole Karakoram region is to show the disparity of DEM observations along stripes, although indeed the area was too large. Here is the new version, with new glacier labels.

**Figure 1.** Main map focusing on the study area in the Karakoram, with regional localisation provided in the inset map. The colour scale shows the number of pre-filtered ASTER-derived elevation observations over the period 2000-2019 from Hugonnet et al. (2021). Glacier outlines from RGI7 are shown in dark tones. The glaciers with the surge events we analyse are outlined in red. The longitude and latitude are expressed in the coordinate reference system EPSG:4326 (WGS84).

**Figure 3**

This is a bit of a nitpick here but I would refrain from using two similar colors for the lines in "Interpolation it. 1" and "Interpolation it. 2" - being colorblind, I can't see the difference between them.

We modified the green line into purple, which is more distinguishable.

In addition, it would be beneficial if you showed the uncertainty associated to each measurement on the plots to the right.

For clarity sake, it is hardly possible to add such information on this figure. We added a new supplement figure for this (Fig. S8).

Finally, I see no mention to any Student-T distribution in the paper (because the methods you rely on make no explicit assumption on the distribution of the data). Rename the "t-interval" into "Confidence Interval".

We modified the legends accordingly, including Fig. 6 and Fig. 9 also, and in the supplement. We let the distinction in the caption, as it is different for the two methods.

**Figure 6**

It's really hard for me to see the individual points between raw elevation and filtered measurements.

I think the symbols and the figure in general are just too small.

We simplified the figure. There is now no more overlaying point, as you can see on the figure below.

Again, just remove t-confidence interval and use confidence interval. i think it's too specific for most readers - if they want to know more, they will read Shekhar et al. (2021).

We modified it accordingly, including Fig. 3 and Fig. 9 also, and in the supplement.

**Figure 7 Increase the size of the figure and individual panels.**

We reduced the size of text at the benefit of the size of panels.

**Figure 12**

Please add the red A, B and C regions in the captions. It's a shame to have to go into the text to grasp what the figure shows.

We completed the caption with "The areas and trends designated in red are discussed in subsection 5.3. They highlight areas of large surge smoothing or removal (zone A) or overall smoothing of elevation changes (trend B) by the original method (Hugonnet et al., 2021), and artefacts created by the presented workflow (zone C)."

**• Table 2**

This table is pretty confusing.

I would suggest replacing Table 2 with a figure showing the distributions for each glacier. This would avoid having 2 columns as the 90th percentile and show the full distribution.

**Figure 11.** Histograms of the elevation difference between the references DEMs and the DEMs of the workflow interpolated at the same dates. We consider only surge-affected areas. Vertical dotted lines are the median of each histogram. The largest median is 5.18 m (resp. -5.63 m) during surge (resp. during quiescence).

---

## Author Response (AR2)

**Response to the comments from the reviewers**

We thank all the reviewers and the editor for their work on our manuscript. Please find below in three different sections (page numbers below) the answer to each reviewer and the editor. Our answers are in blue.

| Answer to the comments from | n William Kochtitzky, reviewer : | #1page 2 |
|-----------------------------|----------------------------------|----------|
| Answer to the comments from | n Mingyang Lv, reviewer #2       | page 3   |
| Answer to the comments from | n Wesley Van Wychen, editor.     | page §   |

**Response to the comments from William Kochtitzky, reviewer #1.**

I appreciate your response, you have developed an excellent manuscript. Congratulation

We thank again Will Kochtitzky for their careful reading of our manuscript and their comments in the whole review process.

**Response to the comments from Mingyang Lv, reviewer #2.**

The authors made all necessary changes to my 1st round comments. They also revised the manu massively according other reviewers' comments. Therefore, I suggest it could be considered for publication after some technical corrections, like:

We thank again Mingyang Lv for their review and feedbacks on our manuscript. We have made all the minor changes suggested, please find below our answers in blue.

**Answers to comments**

- Line 18 & line 36: "important" is not commonly used to describe elevation change during surges.

  These sentences have been removed during the improvement of the whole introduction.
- Line 84: GLO-90 is first used here. Please give more information in this line rather than in the next paragraph.

We replaced the paragraph right after the sentence with the first use. The first sentence of the paragraph have been simplified, as "The Copernicus DEM GLO-90 (European Space Agency and Airbus, 2022) we use as a reference elevation for coarse filtering of very large outliers is edited from data of the TanDEM-X mission acquired between 2011 and 2015."

 Line 266: Wrong use of brackets. Fig. 7d? Also, I suggest to add an sub-figure in top right of Figure 7 and give the centreline of Kyagar Glacier.

We corrected the bracket error.

We added insets of the Kyagar glacier on the main figures. We completed the caption: "The insets for the Kyagar glacier are on the same scale as the main frames.". See below the new figure.

• Line 274: If time series are note presented in the manu or supplement, this sentence may not present in the manu.

We removed the sentence and the mention of this build-up.

About the mass imbalance during surges in the result secssion, the authors only give some
description on Hispar Glacier. If could, please also add some to other four glaciers. Otherwise,
you could add a new section to present the result of mass imbalance during surges.

We deleted the presentation of this volume in the results section. We think that there is tool little to report on this topic to add a new section. We discuss more thoroughly imbalances and volumes in the discussion / comparison with literature section, and let the reader to check at Table 1 for volume details (including imbalance). We added a reference to table 1: "Time series, extracted at regular intervals along the selected centrelines are shown in Supplement (Fig. S3 to S7), and surge volume transfers are reported in Table 1 for each glacier."

**Response to the comments from Wesley Van Wychen, editor.**

Dear authors.

Thank you for your revised manuscript. Although two of the reviewers were quite positive about the changes you have made to the manuscript, after my own review of the manuscript, I believe that there is further work to be done before this manuscript can be considered for publication. Specifically, I find that overall, the structure of the manuscript needs to be improved, the content streamlined and more details about the decisions made in the methodology section need to be robustly described. There are certainly some interesting results presented in this work; however it is hampered by poor articulation and organization in many sections of the manuscript. Content is sometimes presented twice, or at least mentioned twice, and this needs to be addressed. Furthermore, in the methods section, you often note the parameter values chosen in the work but provide little (if any) justification as to why you selected these values. This also needs to be addressed. It may be beneficial to change this manuscript from a research article to a communication. At present, what you are essentially presenting is how this work improves surge characterization from previous elevation change detection work (largely Hugonnet et al., 2021). This is not a problem, but I see this more as an iteration on that work, than something that is wholly separate from that work. Throughout the manuscript, there is a great use of colons to connect thoughts/sentences, but the structure does not really work. Please review the manuscript, check for these instances and revise as needed.

Overall, I believe that there are some interesting results presented in this research, however at present, the content is a bit too muddled for these results to be fully appreciated by the reader. I would encourage the authors to go back through the entire manuscript and in detail revise the entire manuscript for clarity and seek opportunities to shorten content and reduce redundancy in the text. Please also address the lack of details in the methodology section and related to this point, I still believe that the uncertainty could be further elucidated within the text. Below, I have provided an non-exhaustive list of detailed changes that should help with the revision. All the best.

Wesley Van Wychen

We thank Wesley Van Wychen for their positive appreciation of our work and for their detailed review of our manuscript. We tried our best to implement the suggested changes, but we slightly disagree on some aspects of the review. Below we summarize our responses to the general comments:

- manuscript structure that needs to be improved: we re-organized the method section by introducing sub-sections for the different processing steps and we moved the previous section 5.3 "Elevation change comparison" earlier in the discussion. We also added sub-sections to identify each surge event characterized and discussed in the results and discussion sections.
- repetitions: we removed repetitive sentences (mostly from the introduction) and paid particular attention not to present content twice anymore
   Also, we abbreviated the numerous references of "Hugonnet et al. (2021)" to "H21", starting from section 2, except in the figures.
- lack of justification for the choice of the parameters in the methods: we agree that the choices of the filter parameters remain arbitrary, in particular because there is no absolute reference/ground truth to optimize these parameters. However, we think that we made this explicit in the text (e. g., L125-127 and figure 4). We also added a fifth column to figure 4 to show the impact of p and q parameters of the ALPS-REML method.
- suggestion to change from a research article to a communication as our work is seen as an iteration of the work of Hugonnet et al.: We understand that this work may seem like an iteration of the H21 study, as it relies on the same input dataset and with the same goal of obtaining continuous time series, however we disagree that this is a simple "iteration". Both studies differ on several points: 1) H21 focuses on glacier volume change estimates, while our study focuses on surge analysis. Both topics come with different challenges to address. The time series from Hugonnet et al. are not suitable for surge analysis, as shown in multiple places (fig. S1, fig. 6, fig. 7). On the other hand, our goal is not at all to udpate/improve the estimates of H21. 2) The two methods for time series regularization are totally independent, even though they produce the same type of output. It is a real challenge to implement a workflow that is able to filter noisy time series and preserve abrupt changes, which is why we had to come up with a totally new

approach. 3) Both methods can be generalized to other datasets. The application to the same ASTER time series is simply imposed by the lack of other dense elevation times series. We therefore decided to keep our original format but tried to better highlight the novel aspects in the revised manuscript (e.g., we added in conclusion "The workflow, applied to ASTER DEMs but which can be adapted to other datasets, can generate [...]").

- L19: please put surge type in quotation marks
   Done
- L22-23: 'and this subject continues to be the subjects of developments and theories' ◊ this is awkwardly phrase, please have a look and revise.

Rephrased "and are an ongoing focus of theoretical investigations"

L32: This first sentence is not really necessary.

Sentence deleted. This now reads as "Remote sensing analysis from satellite imagery can produce a large amount of digital elevation models (DEMs), providing observations of the elevation of the glacier surface and its variation over time."

• L25-26: you should mention here the different 'states', whether they be mass build up related to quiescence in a reservoir area, mass transfer during a surge etc.

This now reads as "Observations of glacier surface elevation change over time are extremely useful to document glacier surges, and can give insight into the current state of a glacier in its surge cycle. The surge period, active phase of the surge-type glacier, is characterised by thinning (i.e. decrease of surface elevation) in a reservoir area and thickening in a receiving area, representative of the ongoing mass transfer. The quiescence period consists in strong thinning in the receiving area of the previous surge, and a thickness increase (mass build-up) before the next surge and mostly in the future reservoir area. The differencing of elevation maps permits one to compute the volume of ice transferred during a surge event and determine the spatial extent affected (ref)."

• L26: 'it permits one to'

Done

L27: 'along with' change to 'and determine'

 L28: an information --> information Done

• L31: at 'Remote sensing...' this should be the start of a new paragraph.

L31: 'permits one to'

Replaced by a change relative to a previous comment.

L33: along time --> over time
 Done

• L36: Surges are short-term events --> This depends on how you define short, we have observations that surges may last decades, see examples in the Canadian Arctic.

This sentence has been modified, it now reads as "the use of DEMs for the study of surges is often limited to a few dates or specific case studies, because the temporal availability of DEMs does not always match the surge phases. The retrieval of mass transfer variations happening during such surge events requires [...]"

• L40-41: end of sentence, change to 'long enough to capture a number of surge events in their entirety'.

Rephrased to "long enough to capture a number of surge events and a few complete surge cycles."

L41; 'altimetry missions'

L41: such as ICESat-2, CryoSat-2 etc. --> are there really many more than this to warrant etc?

L42: but 'their spatial sparsity' change to 'their limited spatial sampling'

L43: 'several studies have used dense SAR time series on surge case studies, usually without time series filtering technique' --> this is unclear. Do they use DEMs derived from SAR data to do the analysis or backscatter analysis. At the moment, it is not clear how this statement connects to the previous sentences.

We removed this section of the introduction, which was a bit off-topic.

- L48-49: revise to 'elevation precisions of a similar magnitude'. Also, here do you mean elevation precision of 15 m?
- L49: and sometimes large artefacts caused by clouds, jitter, lack of stereo correlation on saturated/textureless terrain.

This part have been reworked for clarity.

As ASTER is the only systematic optical stereo sensor, we are now specific to this sensor and not broad optical sensors: "The DEMs derived from ASTER have an elevation precision of about 5-20 metres and they can have large artefacts caused by cloud sensitivity, satellite jitter or lack of stereo correlation on saturated/textureless terrain (Berthier et al., 2023; Girod et al., 2017)"

• L51-52: these sentences can be combined and should be revised a bit.

This now reads as "Such noisy DEM time series require specific filtering techniques that preserve surge signals (i.e., preserve elevation observations before, during and after the surge), as basic thresholds and linear methods used to assess long-term elevation changes might misinterpret surge observations as outliers."

• L52: Also, the volume transported or slope should be computed at consistent dates across a whole glacier. --> for what reason? This is a bit of a sentence fragment at the moment.

This should be better: "Furthermore, the estimate of volume transported and the surface slope are sensitive to data gaps and their interpolation. As a consequence, the need to be computed at similar dates across a whole glacier to ensure physical consistency."

L53: At 'Various' start a new paragraph

Done

L61-62: sentence needs more revision here.

Done

- L65: In this study we aim at developing --> In this study we develop a workflow Done
- L66 dataset or datasets?

Done

• L68-69: "We apply it to an ASTER DEM dataset from Hugonnet et al. (2021). We produce a regional dataset in the Karakoram region covering more than 100 surge-type glaciers." ◊ "We apply it to an ASTER DEM dataset from Hugonnet et al. (2021) to produce a regional dataset in the Karakoram region covering more than 100 surge-type glaciers.'

Done

L70: 'We also compare the surge characteristics such as volumes transferred to other products and studies.' Please list some of the other products and studies here'

This now reads as "We evaluate the performance of the workflow compared to the results of Hugonnet et al. (2021) and other DEMs (SPOT and HMADEM). We also compare the surge characteristics such as timing and volumes transferred to other studies (e.g., Bhambri et al., 2022; Steiner et al., 2018; Gao et al., 2024)."

- L75: 8. 6% I assume this is meant to be 8.6%?
- L76: Guillet et al. (2022) identified 223 surge-type glaciers among glaciers larger than 5 km2 (not individualizing tributaries) --> Guillet et al. (2022) identified 223 surge-type glaciers larger than 5 km2 not individualizing tributaries.

Done

- L76-77: these studies or this study? Are you referring to both studies above in this paragraph, or
  just the Guillet et al., (2022) in this instance. If just Guillet et al., (2022), then revise to this study.

  Done
- L78-79: We use the DEMs produced in the global study of Hugonnet et al. (2021), which ranged from 07/2000 to 09/2019 in the Karakoram. They are generated from satellite images of the ASTER sensor. Revise to: 'We use the DEMs produced in the global study of Hugonnet et al. (2021), which ranged from 07/2000 to 09/2019 in the Karakoram and were generated from satellite images from the ASTER sensor.'

Done

L84/ L93-94: Can the information about the Copernicus DEM all be presented at the same time that it is mentioned at L84? Also, please revise these sentences, there are some fragments here that could be improved. Here, you also discuss the impact of radar penetration, but no context is given about this topic, please add a sentence to provide some context here about why radar penetration is a factor to consider.

Paragraph moved accordingly to a comment from reviewer 2. The last sentence now reads as "The impact of radar penetration in ice and snow (up to about 10 metres) creating a bias in TanDEM-X elevation estimate is negligible compared to the threshold we use (hundreds of metres) (Berthier et al., 2023; Rizzoli et al., 2017)"

L96: 'We aim to develop' --> this statement is a bit tentative, you either have or you haven't developed this. So, I would suggest that you change this to: "Here we present a workflow to ...."

Done

L95-100: So, is the major difference here in the methodology, just your pre-processing steps from what is presented in Hugonnet et al., (2021)?

We clarified a bit this: "We use the ASTER DEMs of Hugonnet et al. (2021), but processed them with a different workflow, because Hugonnet et al. (2021) workflow performs weakly on surge events (see for example figure S1). We use Hugonnet et al. (2021) workflow as a baseline to compare our own workflow to highlight the improvements for the surge cases.". The major difference between our study and Hugonnet et al. (2021) is in our entire workflow: use of LOWESS algorithm versus linear filters, ALPS-REML interpolation versus Gaussian Process Regression...among others. The so-called "pre-processing" step is a minor adaptation step mostly specific to this dataset which are not relevant for other dataset (including ASTER DEMs generated differently) and which we chose to not write in our workflow in this paper to highlight the core of the workflow and make the workflow more clearly applicable to other dataset than the one we use.

L101: There is some redundancy in this first bullet point.

This is clearer now: "LOWESS workflow, core step of the filtering: we apply an iteration of the LOWESS algorithm"

• L109: Here you mention that the LOWESS sequence will be described in detail later, but you have already mentioned earlier in the manuscript that it will be described in subsection 3.2. This should be revised and harmonized.

Done

L110: There are quite a few fragments here, that could be revised and included in a single sentence.

This now reads as "We filter the elevation time series by two iterations of the non-parametric LOWESS algorithm, which is a moving weighted regression"

• L115: can you provide more detail on why you selected the 0.4 and 0.3 values? This really is not detailed sufficiently enough.

L116: again, please provide a rationale for the choice of 2.

L117-118: again, please provide greater rationale for the choice of 'symettric'

We agree that there are a number of parameters that were manually chosen, which might appear a bit arbitrary to the readers. We tried to make this statement more explicit in the revised manuscript: "Here are the main parameters set for each LOWESS iteration. They have been manually tuned after visual evaluation on a number of time series samples, both with and without surge signal (Fig. 4). We caution that these parameters were chosen specifically for the ASTER DEM dataset and might not all be suitable for other datasets (discussed in subsection 5.4)."

L128: what constitutes 'worse' in this case?

We tried to improve the phrasing: "Time series with both large temporal data gaps and a noisy signal can create computational errors for small smoothing parameters."

L133: abbreviation needs to come after the fully detailed title.
 Done

L138: drive what over-fitting behaviour? Any over-fitting behaviour?

We now stress that over-fitting is particularly a problem for noisy data like ASTER DEMs: "[...], thus allowing us to narrow down the effect of the regularization/smoothing specifically on the high-frequency components. The latter are responsible for the over-fitting behaviour of the model, i.e., the fact that it fits too closely or exactly to the training data and becomes inaccurate for new data. This is particularly problematic for noisy time series like ASTER DEMs."

- L142: GCV metric..... --> this sentence is awkward and is in need of some revision.
   Done: "The GCV metric quantifies the generalization error of the model by making predictions at data points that were not used to fit the regression model."
- L150: The same ALPS-REML code as what? Again, please provide more detailed rationalization of the parameters you set.

For more transparency on the choice of parameter values, we added to Fig. 4 the plot of ALPS parametrization. Here is the new figure and caption.

Figure 4. Impacts of the different LOWESS parameters (lines 1 to 4) and ALPS parameters (lines 5 and 6) on the regression/interpolation solutions. Plain lines are the final selected values. The line corresponds to three different data points (TSa-c in the column order, locations shown on Fig. 7.c).

- L189: in metres?
   Cubic metres, added to the sentence.
- L 191: elevation change uncertainty --> but what is this value? It seems under described this far
  in the manuscript.

We removed the whole sentence; such justification is not expected here.

- L230: fewer points?
   We deleted this second part of the sentence; it is unnecessary.
- L255: in Figure 6.a2 Done.
- L302: wuth --> with Done.

• In the results section, please create subsections for each individual glacier surge event: Hispar, Khurdopin, Kyagar and Yazghil, this will help with organization of the text.

We added subsubsection for each case study, both the result section and the discussion section.

- L330: uncertainty representation. This should be moved to the methods section, it also seems
  confusing that this is mentioned after the reader would have accessed these figures.

  Done.
- L337-339: is it possible that areas of large difference is simply due to imperfect comparisons made in areas of very steep terrain where we might expect large differences over short distances?

This might sometimes be possible over limited areas, yes. However, in this case, we identify several cases on glacier areas that are not steep (e.g. Fig. 10.b at the Kunyang-Hispar junction, even though ASTER elevations are consistent with each others along time at this location and period). For another example, we also identify a SPOT DEM with a clear and large artefact of several tens of meters on a glacier tongue (not presented in the paper).

L338-339: Please revise this sentence, how is a wrong estimate credible?
 Rephrased to "can occur", meaning it is possible.

L340: What is meant by moderate? Please be more quantitative in the description.

This now reads as "The map of elevation differences on the glaciers shows differences of a few meters overall, which is moderate compared to the elevation change amplitude of the surge (Fig. 10). The difference may be important such as several tens of metres locally at the surge front (e.g., Fig. 10.a-b at the Kunyang-Hispar juction)."

- L341: is of -4.3 m --> is -4.3 m Done.
- L342: 2015-10-13, -5.2 m --> this section of the sentence needs revision.

We rephrased as: "Across the entire glacier areas, consistent discrepancies are observed. For instance, on 2015-10-13, the Hispar glacier exhibited a median difference of -4.3 m with a standard deviation of 9.7 m. Similarly, on 2015-11-28, the Braldu glacier exhibited a median difference of -5.2 m with a standard deviation of 8.7 m."

• L350-353: end of paragraph needs revision for clarity.

This now reads as "In such situations, our method of filtering and interpolation usually leads to an underestimate of the transferred volume and an overestimate of the surge duration (e.g., twice its duration for Kyagar glacier). Onset and end dates cannot be determined precisely between two observations separated by more than 6 months or a year, even on filtered series, as it occurs for the Kyagar glacier"

L355: why 450 days? This should be specified.

The duration of 450 days is arbitrary but realistic, as it is rather common to observe intervals with more than 400 to 500 days without observations after filtering. We added two sentences:: "Each iteration results in a period of at least 450 days without observation, which is common in the filtered series. For instance, on the surge-affected area of the Kyagar glacier which is subject to a lack of observation for our processing, there is on average 3 time intervals of 400-to-500 days per time series (1 time intervals for the Hispar glacier, in comparison)."

- L358: larger for larger --> larger for longer Done.
- L359: Case b) is specific --> it is unclear what is being referred to here.

Done.

 L370: may not permit to observe this phenomena --> may not allow this phenomena to be observed

Done.

L381-383: this was discussed earlier in the manuscript, correct? Why not just make sure that
everything is described once collectively.

L289-291: Okay, but you have a difference of 687 +/- 414 x 106 m3, so what happened to this ice?

We removed the sentences about volumes from the result section. They are now only presented in the discussion, section 5.2.

L386-387: it is not clear to me how the reader is meant to understand the information in this sentence.

This now reads as "Another possible source of imbalance is the impact of crevasse opening during the surge, which can represent a non-negligible volume change. As an example, the opening of crevasses can be equivalent to up to 0.2 m thickness at regional scale of the Greenland Ice Sheet (Chudley et al., 2025). As inland parts of these regions are largely crevasse-free, we can expect such impact on the volume to be significantly larger over the highly crevassed post-surge surface of Hispar glacier."

- L402: period misplaced after (2018).
   Done.
- L404: Is the information here from the Steiner et al., (2018) or from the personal communication?

We do not write the previous volume estimate anymore, only the one reassessed from the personal communication. We removed the citation of the original publication to avoid confusion. This now reads as "The surge started in October 2016 according to Imran and Ahmad (2021), about 7 months later than our estimate (Table 1), and late August 2015 according to Steiner et al. (2018). The volume received in the receiving area is estimated at 1182 x106 m3 during late August 2015 (elevation extrapolated linearly from TanDEM-X in 2011) to May 2017 (ASTER) data (Jakob Steiner, personal communication)."